# Learning Place Cell Representations and Context-Dependent Remapping

**Markus Pettersen**
Department of Numerical Analysis and Scientific Computing
Simula Research Laboratory
Oslo, Kristian Augusts Gate 23
`markusb@simula.no`

**Frederik Rogge**
Department of Biosciences
University of Oslo
Oslo, Blinderveien 31
`frederik.rogge@ibv.uio.no`

**Mikkel Elle Lepperød**
Department of Numerical Analysis and Scientific Computing
Simula Research Laboratory
Oslo, Kristian Augusts Gate 23
`mikkel@simula.no`

## Abstract

Hippocampal place cells are known for their spatially selective firing patterns, which has led to the suggestion that they encode an animal's location. However, place cells also respond to contextual cues, such as smell. Furthermore, they have the ability to remap, wherein the firing fields and rates of cells change in response to changes in the environment. How place cell responses emerge, and how these representations remap is not fully understood. In this work, we propose a similarity-based objective function that translates proximity in space, to proximity in representation. We show that a neural network trained to minimize the proposed objective learns place-like representations. We also show that the proposed objective is easily extended to include other sources of information, such as context information, in the same way. When trained to encode multiple contexts, networks learn distinct representations, exhibiting remapping behaviors between contexts. The proposed objective is invariant to orthogonal transformations. Such transformations of the original trained representation (e.g. rotations), therefore yield new representations distinct from the original, without explicit relearning, akin to remapping. Our findings shed new light on the formation and encoding properties of place cells, and also demonstrate an interesting case of representational reuse.

## 1 Introduction

Animals and humans are capable of extraordinary feats of navigation, from birds migrating by following magnetic fields [Packmor et al., 2021], to rodents navigating mazes [O'Keefe, 1976, Small, 1901] and cab drivers memorizing and traversing nearly 26000 busy London streets [Fernandez-Velasco and Spiers, 2023]. In the brain, navigation ability is believed to be supported by Hippocampal place cells [O'Keefe and Dostrovsky, 1971, O'Keefe and Nadel, 1978]. Place cells are known for

38th Conference on Neural Information Processing Systems (NeurIPS 2024).

their tendency to only fire at one, or a few locations within a recording environment [Park et al., 2011], correlating with the position of the animal. Besides encoding allocentric location, place cells also respond to contextual cues, such as room identity, room geometry, odors or colors [Latuske et al., 2018, O'Keefe and Burgess, 1996, Leutgeb et al., 2005b, Jeffery, 2011]. In other words, place cells form conjunctive representations, merging spatial information with available contextual cues. It is also believed that place cells distinguish contextual information trough so-called remapping. In response to large changes in the environment, place cell responses modulate, and spatial representations can become uncorrelated across contexts [Muller and Kubie, 1987, Leutgeb et al., 2004, 2005b].

How place cells obtain their striking behaviors remains a matter of debate. Some argue that place cells inherit their firing fields from upstream cell types, such as grid cells [Hafting et al., 2005, Solstad et al., 2006, Jeffery, 2011], or border cells [Barry et al., 2006, Pettersen et al., 2024] or that place cell representations in different environments form distinct attractor states [Jeffery, 2011, Samsonovich and McNaughton, 1997]. However, exactly how place-like representations emerge, and how they can be learned, remains poorly understood. In recent related work, however, a range of normative models have demonstrated that neural networks trained to solve simple navigation tasks actually learn representations similar to biological spatial cells [Cueva and Wei, 2018, Banino et al., 2018, Uria et al., 2020, Sorscher et al., 2022, Whittington et al., 2020, Xu et al., 2022, Dorrell et al., 2022, Schaeffer et al., 2023]. However, these models often feature complicated architectures and a range of different regularization strategies, making it difficult to discern why the observed representations actually arise. Furthermore, some of these works tend to focus on learning grid cell-like representations, placing less emphasis on other emergent cell types, such as place cells.

In this work, we take inspiration from existing machine learning models [Cueva and Wei, 2018, Banino et al., 2018, Sorscher et al., 2022, Xu et al., 2022, Dorrell et al., 2022] and propose a minimal, similarity-based objective. When a feedforward network is trained to minimize this objective, we find that it learns place-like spatial representations. We further show that the objective is easily extended to encompass joint representations of space and context. When trained in a joint setting across multiple contexts, we find that the network learns uncorrelated representations when comparing different contexts, similar to Hippocampal global remapping. We further train a recurrent neural network to solve the same task, and find that band cell-like [Krupic et al., 2012] representations emerge alongside the place code, showing that other cell types may be involved in path integration, and also that the objective extends to more naturalistic settings. Lastly, we show that we can apply orthogonal transformations to learned spatial representations in order to generate new spatial maps while preserving the similarity structure. Thus, the proposed objective allows for switching between different maps without having to relearn them, offering an interesting perspective on Hippocampal remapping.

## 2 Results & discussion

### 2.1 A similarity-based objective for learning representations of space and context

We consider the problem of learning an encoding of some region of space (e.g. a square recording enclosure). We follow the example set by recent normative models [Dorrell et al., 2022, Schaeffer et al., 2023], and argue that biologically plausible spatial representations can be obtained directly by specifying the properties of the network population vector. Considering properties that a spatial representation should have, we propose an objective where we demand that: 1) Points that are close in physical space should be represented by similar population vectors. 2) Points that are distant in physical space should be represented by dissimilar population vectors. 3) In the open field, no point or direction is special, so the above properties should be rotation- and translation invariant in space. 4) Unit activations are bounded. 5) Unit activations are non-negative.

To investigate representations with these properties, we train neural networks to minimize a spatial encoding objective. Consider a neural network with population vector $\mathbf{p}(\mathbf{z}_t)$, where each component $p_i(\mathbf{z}_t), i = 1, 2, 3, ..., n_p$ is the firing rate of a particular (output) unit of the network at a particular location, $\mathbf{x}_t$ (where t indexes e.g. time along a trajectory or sampled spatial locations). $\mathbf{z}_t$, on the other hand, is the network's latent position estimate corresponding to $\mathbf{x}_t$. We impose non-negativity in the architecture of the neural network by selecting appropriate activation functions. With these

prerequisites in mind, we explore the following objective function

$$\mathcal{L} = \mathbb{E}_{t,t'} \left[ \left( \beta + (1-\beta)e^{-\frac{1}{2\sigma^2}|\mathbf{x}_t - \mathbf{x}_{t'}|^2} - e^{-|\mathbf{p}(\mathbf{z}_t) - \mathbf{p}(\mathbf{z}_{t'})|^2} \right)^2 + \lambda|\mathbf{p}(\mathbf{z}_t)|^2 \right], \qquad (1)$$

where $|\cdot|$ denotes the L2-norm, while $\beta$ is a lower bound on the target similarity, $\sigma$ is a hyperparameter that controls the scale of the learned similarity structure, and $\lambda$ a hyperparameter governing an L2 activity regularization term.

Intuitively, (1) compares the (Gaussian) similarity of two points $\mathbf{x}_t$ and $\mathbf{x}_{t'}$, and asserts that the corresponding population vectors at those points ($\mathbf{p}(\mathbf{z}_t)$ and $\mathbf{p}(\mathbf{z}_{t'})$) should exhibit the same similarity. In other words: Points that are close in physical space should be represented similarly, while distant points in physical space should be represented using dissimilar population vectors. This idea is illustrated in Fig. 1a). $\beta$, on the other hand, determines the similarity scale at which vectors are deemed dissimilar. This borrows from the concept of vectors being *nearly* orthogonal in hyperdimensional computing [Kanerva, 2009], which exploits that (random) vectors in higher dimensions tend to lie some intermediate distance from other vectors. For the higher-dimensional population vectors of neural networks, dissimilarity may therefore be meaningfully defined in terms of some intermediate, rather than zero similarity (and maximal population vector separation). See App. B for more details on the influence of $\beta$ on learned representations.

We also find that the objective in (1) is just a special case of a more general encoding objective, where distinct sources of nonspatial information can be encoded in a single population vector. Fig. 1b) illustrates how contextual signals can be represented in a similar manner to the spatial case. However, place cells do not encode contextual information exclusively, but rather respond to particular locations and contexts in conjunction. Again, the similarity objective can be extended to accomodate this. If we represent context information in the simplest manner, as a scalar signal $c$, we can have the neural network encode spatial and contextual information jointly by training it to minimize

$$\mathcal{L} = \mathbb{E}_{t,t'} \left[ \left( \beta + (1-\beta)e^{-\frac{1}{2\sigma_\mathbf{x}^2}|\mathbf{x}_t - \mathbf{x}_{t'}|^2 - \frac{1}{2\sigma_c^2}(c_t - c_{t'})^2} - e^{-|\mathbf{p}(\mathbf{z}_t, c_t) - \mathbf{p}(\mathbf{z}_{t'}, c_{t'})|^2} \right)^2 + \lambda|\mathbf{p}(\mathbf{z}_t, c_t)|^2 \right],$$
$$(2)$$

where, in general $\sigma_\mathbf{x}$ and $\sigma_c$ are spatial and contextual encoding scales, respectively. For this work, however, we set $\sigma_\mathbf{x} = \sigma_c$. As with the purely spatial and purely contextual case, we model distinct spatial/contextual combinations as similar or dissimilar population vectors. An illustration of this situation is provided in Fig. 1c). When trained to minimize (1) or (2), both path integrating recurrent and position-encoding feedforward units learns place-like representations, as shown in Fig. 1d). Due to the correspondence with recurrent representations, their simplicity and ease of training, we employ feedforward networks for most analyses in this work. However, we show that our findings extend to recurrent networks, and that these learn additional path integration. We also train recurrent networks in a more biologically plausible manner, to demonstrate that our results hold in more naturalistic settings (see App. A).

## 2.2 Feedforward networks learn place cell-like representations

Having shown that place-like representations emerge in networks trained to minimize (1), we now study the feedforward network and the influence of model hyperparameters in more detail. Importantly, networks are able to minimize the objective function for a large range of hyperparameter combinations (evidenced by saturated loss in Fig. 2a)), indicating that models are capable of learning the desired similarity structure. This is also reflected in Fig. 2f), which shows an example of the agreement between the desired spatial similarity, and the corresponding learned representational similarity at the center of the training environment.

Depending on the choice of model parameters, the learned representations exhibit several features observed in biological place cells. While most units display strong tuning to particular spatial locations (see Fig. 2b) for several examples), activity levels vary, with several units being completely silent (Fig. 2b), c) and d)), similar to e.g. [Thompson and Best, 1989, Alme et al., 2014]. For small values of $\sigma$ and nonzero $\beta$, some units exhibit multiple place fields (e.g. Fig. 2b), left), which is also observed in biological cells [Park et al., 2011], especially for large spaces [Harland et al., 2021]. We also observe that field locations cover the training arena, shown in Fig. 2g), enabling encoding of the entire environment.

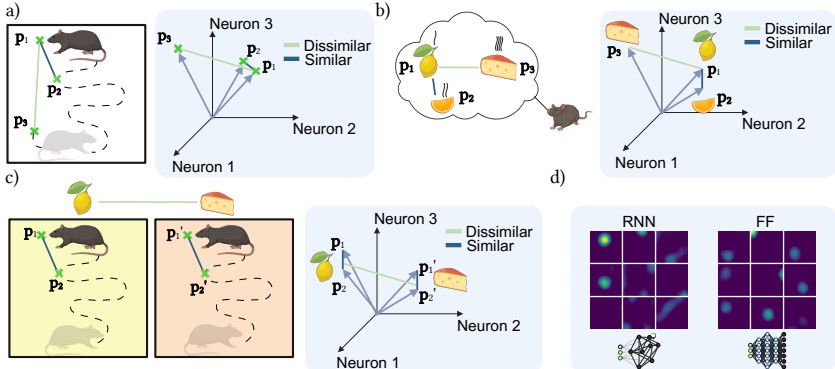

Figure 1: **Overview of models and objective**. a) Illustration of the spatial objective: locations that are close should be encoded by similar population vectors, distant locations by dissimilar population vectors. b) Similar to a); similar context signals should be represented by similar population vectors, dissimilar contexts by dissimilar population vectors. c) Similar to a) and b), but for joint encoding of space and context. d) Ratemaps of randomly selected units in networks trained to minimize the spatial objective function. Shown are learned representations for a recurrent network performing simultaneous path integration, and a feedforward network performing spatial encoding.

We find that the properties of the learned representations follow readily from model hyperparameters. For example, Fig. 2b), d) and e) demonstrates that unit field size increases with increasing scale parameter $\sigma$, and that mean rates increase accordingly. Also, the number of place fields for a given unit is strongly linked to the similarity lower bound $\beta$ (see App. B for more on the influence of different hyperparameters).

Next, we investigated whether the learned representations in our model constitute useful spatial representations, in the sense of being decodable. First, we decoded the position using a weighted mean of peak locations based on (5), varying the number of units included in the analysis. The results, shown in Fig. 2h) indicate that the decoding performance (measured by the mean Euclidean distance between the actual and decoded positions) differs based on the number of units utilized, and model parameters, and demonstrate that the peak locations of learned representations can be used for accurate position decoding (achieving best-case mean error of a few percentage points relative to the arena size). Fig. 2 i) and j) show a comparison of all three methods. Interestingly, a simple linear decoder performs best in terms of mean error, while the population decoding using uniformly distributed memories exhibits the lowest maximum error. Moreover, the errors are uniformly distributed over space, while for the linear decoder and the Top n decoding we find areas of higher errors, especially at the edges (see Fig. 2 k)). Together, these findings show that the learned representations can be used to decode position efficiently using fairly simple decoding schemes.

### 2.3 Feedforward networks learn global-type remapping

When trained to encode multiple contexts in conjunction with spatial location, feedforward units exhibit dramatic firing field shifts, when comparing across contexts. This effect is shown in Fig. 3a), where ratemaps of randomly selected units are tracked while varying the context signal.

Notably, firing fields exhibit several place-like behaviors, such as multiple firing fields [Park et al., 2011] (e.g. row seven), and field shifts between contexts, i.e. remapping [Muller and Kubie, 1987]. In our case, the observed remapping is not a sudden, attractor-like shift, which has been observed in place cells [Wills et al., 2005], but rather a gradual transition between representations, which has also been seen during Hippocampal remapping [Leutgeb et al., 2005a]. However, we find that spatial representations are uncorrelated between dissimilar contexts, an example of which is shown in Fig. 3b). In fact, for sufficiently different contexts, the median spatial correlation between active units tends to zero, which aligns with global remapping in biological place cells [Leutgeb et al., 2004] (see Fig. 3c) and d)). Spatial correlations in Fig. 3c) also support the observation that units remap gradually for smooth context transitions, as correlations decay away from the diagonal.

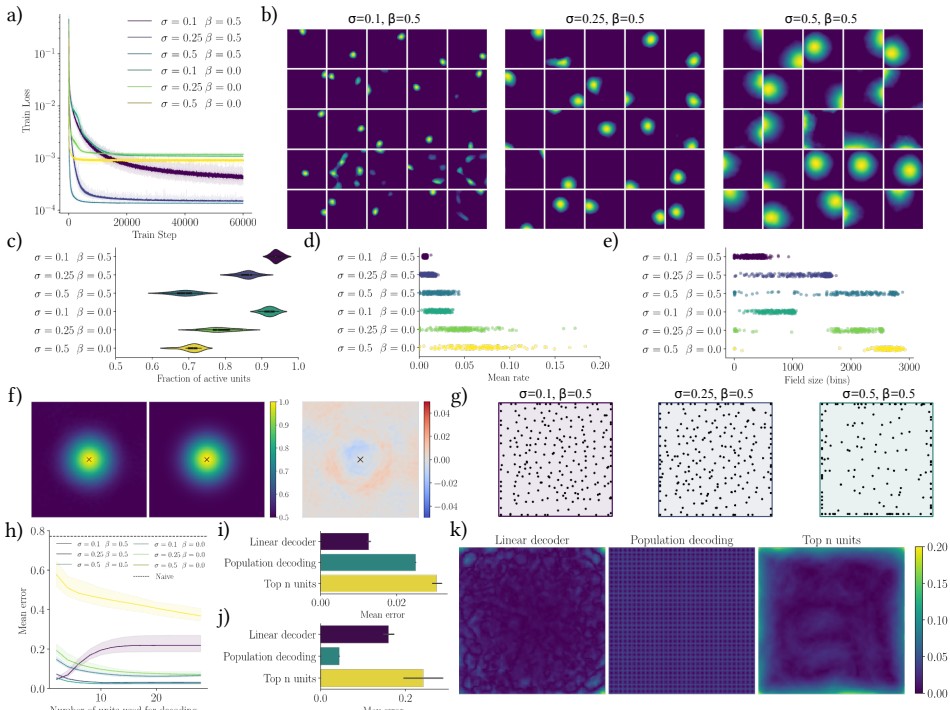

Figure 2: **Feedforward network results.** a) Training loss for different parameter combinations. Line shows the mean of 10 models and error bands show the min and max across models. Note that training data is generated continuously. b) Example ratemaps of randomly selected active units for models with different scale parameters $\sigma$. Color scale is relative to individual unit activity. c) Distributions of the proportion of active units (mean rate $> 0$) for different parameter combinations across 10 models. d) Distribution of mean rate of units for each parameter combination (shown for one example model each). e) Field sizes in pixels for each parameter combination (shown for one example model each). f) Left: Example target similarity structure relative to center. Middle: corresponding similarity for the learned representations of model with $\sigma = 0.25$ and $\beta = 0.5$. Right: difference between target and learned similarity. g) Peak locations of all units for different parameter combinations (shown for one example model each). h) Mean position decoding error as a function of the number of units used for Top $n$ decoding. Dashed line shows the naive case where every decoded position is at the center. i) and j) Mean and max decoding error for different decoding methods for trained 10 models, each with $\sigma = 0.25$ and $\beta = 0.5$. k) Example decoding error maps for different decoding methods ($\sigma = 0.25$ and $\beta = 0.5$).

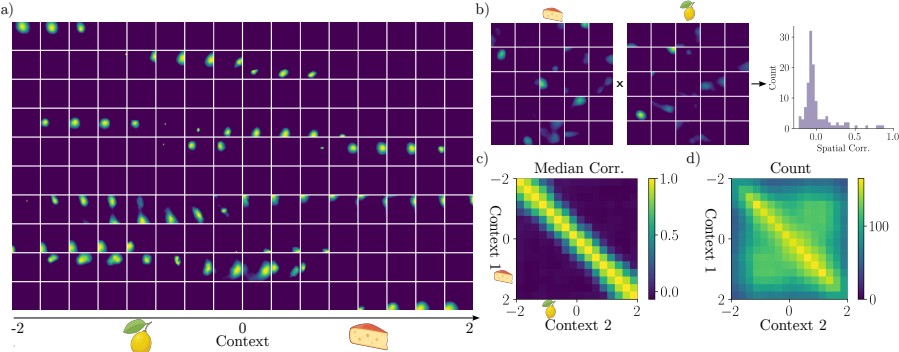

Figure 3: **Feedforward network remapping results**. a) Ratemaps as a function of context, for a random selection of 10 units. Each row corresponds to one unit and each column to a particular context value. b) Example distribution of spatial correlations for ratemaps corresponding to two distinct contexts (context 1 =-0.9 , context 2 = 1.2). c) Median spatial correlations when comparing across all contexts. d) Number of units included (units active in both contexts) in the analysis in c).

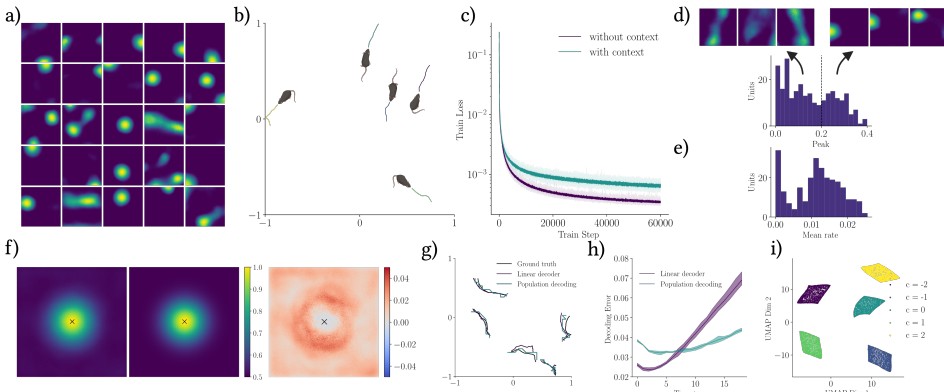

Figure 4: **Recurrent network results with and without context**. a) Ratemap examples of randomly selected units of a recurrent network without context. b) Example trajectories used for training. c) Training loss for recurrent networks with and without context (10 models each, error bands show min and max). d) Histogram of peak values of a recurrent network without context and example ratemaps of units of different parts of the distribution. e) Histogram of mean rates of a recurrent network without context. f) Similarity structure in the center location of the learned representations of a recurrent network without context (left) and the objective (center), as well as the difference between the two (right). g) Example trajectories decoded from network representations h) Comparison of the mean decoding error using a linear decoder or population decoding across trajectories for 10 different models each. i) 2D UMAP projection of spatial representations for different contexts.

## 2.4 Recurrent networks learn place- and band-like representations and path integration

When we train a recurrent network to solve the proposed objective functions (1) and (2) while path integrating along simulated trajectories, we find that its units learn place-like and band-like [Krupic et al., 2012] spatial tuning. Example ratemaps and trajectories are shown in Fig. 4a) and b), respectively. We also find that the recurrent network performs on par with the feedforward network, achieving similar loss minima, both for spatial and joint encoding (see Fig. 4c), suggesting that the network has learned to path integrate and minimize the objective. As band-like representations emerge only in the recurrent network, we speculate that these representations may be involved in path integration, which has also been found in other neural network models [Schøyen et al., 2023]. Also worth noting is that while mean rates are similar, peak rates are markedly different between unit types (Fig. 4d) and e)), further hinting at different functional roles. As with the feedforward model, recurrent responses accurately capture the desired similarity structure (Fig. 4f)).

To verify that the RNN is path integrating, we use both population and linear decoding schemes to extract position estimates from the network representation (Fig. 4g) and h)). We find that positions can be accurately decoded using both methods for sequences longer than the network training trajectory length (see Fig. 4g) for example decoded trajectories). However, the trainable linear decoder is more performant during early timesteps, but exhibits a larger error over trajectory time, suggesting that the population decoding scheme can decode locations more robustly.

Besides simply being able to encode locations and contexts, we find that the RNN learns low-dimensional structures representing distinct contexts. After applying UMAP to recurrent representations in different contexts, we observe that low-dimensional projections capture both the geometry of the square enclosure, and the identity of the context (Fig. 4i)). It therefore appears that the network has learned veritable cognitive maps [O'Keefe and Nadel, 1978] of different contexts, accessible from the representations themselves. See App. D for recurrent units representations across contexts.

## 2.5 Reuse by orthogonal transformation

Having demonstrated that networks learn distinct representations of different contexts, we turn to an interesting feature of the similarity objective. Assuming that we have a trained (feedforward) network, with representations $\mathbf{p}(\mathbf{x}_t)$ (for all $\mathbf{x}_t$ in the domain), the loss function only depends on the norm of, and distance between population vectors (and data). Thus, the objective is invariant

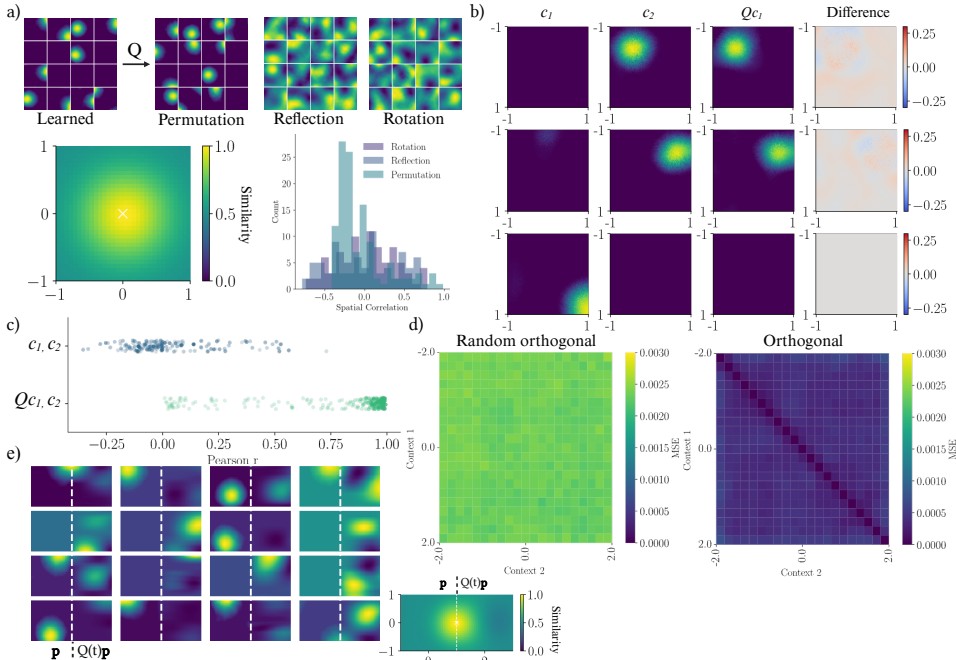

Figure 5: **Remapping by orthogonal transformations**. a) Random global orthogonal transformations (reflection, rotation, and permuation) applied to a trained representation (top) all preserve the similarity objective (bottom left), while producing spatially decorrelated representations (bottom right). b) Best-fit orthogonal transformations applied to learned representations of a feedforward network across two contexts. Inset is the original representation, the orthogonally transformed representation, and the secondary representation alongside the difference between the two for example units. c) Jitter plot of Pearson correlation between ratemaps across contexts for the transformation in b); shading indicates mean unit activity. d) Mean squared error between transformed and original representations for random and best-fit orthogonal transforms across all learned contexts. e) Ratemaps of units where a learned representation (left of dashed line) is extended by a continous orthogonal representation into a novel representation (right of dashed line) without learning. Inset is the corresponding similarity structure, measured from the center of the enlarged environment.

to an orthogonal transformation $Q$ of the representation (and its inputs). In other words, we can transform the entire set of population vectors (using a global orthogonal transform), and still have a representation that minimizes (2). We demonstrate this in Fig. 5a) which shows how different global (and random) orthogonal transformations can be used to produce new representations, that preserve the similarity structure but exhibit low spatial correlation with the original representation, similar to (global) remapping. In particular, we observe that permutations induce more strongly decorrelated representations than reflections or rotations.

Furthermore, we find that global orthogonal transformations can be used to explain some of the representational changes learned by the network through training. Specifically, we computed global orthogonal transformations to match spatial representations across different learned contexts (see 4.3 for a description). As an example, Fig. 5b) shows unit ratemaps in two dissimilar contexts, alongside ratemaps (and ratemap differences) for an orthogonal transformation taking context 1 into context 2. Notably, learned and transformed ratemaps are highly aligned. This is also shown in Fig. 5c), which demonstrates that transformed and learned representations are highly correlated. Furthermore, Fig. 5d) shows that best-fit orthogonal transformations achieve low errors (substantially lower than a random orthogonal baseline) across all learned contexts, suggesting that the orthogonal transformations can account for much of the learned remapping behaviors of the network.

Finally, we demonstrate that continuously applied orthogonal transformations can even be used to extend existing learned representations into novel ones. An example of this is shown in Fig. 5e), where population vectors of a trained representation is transformed along the horizontal axis into

a previously unseen region without explicit training. Notably, extended representations appear to maintain their place-like tuning, and approximately adhere to the original similarity structure. Note that the extended representations are no longer necessarily non-negative (see App. C for details).

As a result, orthogonal transformations could prove a viable way of modeling Hippocampal remapping, and possibly field formation itself. Our findings could also be extended to study what kind of upstream representations are needed to induce orthogonal transformations in Hippocampal representations. Doing so could conceivably shed light on interactions between place cells and other cell types during remapping, in which other spatial cells such as grid cells are often implicated [Latuske et al., 2018].

## 2.6 Limitations

While our model provides a fresh perspective on the formation and remapping of place cells, there are several factors that limit its scope. For one, we consider models where label locations and pre-processed context signals are available during training time, which could be biologically implausible (but see App. A for a recurrent model with more plausible inputs). A related concern is the use of a scalar context signal. However, our model could accommodate more complex context signals, such as spatially bound contexts, or contexts with multiple features. Extending the context signal in this way could prove to be an interesting avenue of research. Another limitation is the use of rate coding; it is somewhat unclear how the proposed objective could be extended to spiking networks. A third limitation is the choice of the similarity measure and its lower bound, i.e. Gaussian similarity with a particular $\beta$. However, this could conceivably be addressed by exploring representational similarity in experimental data in the future.

# 3 Conclusion

This work introduces a similarity-based objective to explain the functional characteristics of place cells. Using this minimal self-supervised objective, we are able to directly demonstrate how place cell-like representations can be learned, and how they can be understood as translating similarity in location to similarity in representation. Furthermore, we observe emergent global remapping as a consequence of joint encoding of space and contexts. Finally, we demonstrate that remapping may be enacted through orthogonal transformations without explicit relearning. By demonstrating how place-like representations can be constructed to encode both space and contexts our findings contribute to a deeper understanding of the neural basis of navigation and memory.

# 4 Methods

## 4.1 Models and training details

We trained two distinct networks to minimize the proposed objective functions (1) and (2): a feedforward network and a recurrent network, as illustrated in Fig. 1d). Models were implemented and trained using the Pytorch library [Paszke et al., 2019]. The feedforward network featured two densely connected layers with 64 and 128 hidden units, followed by an output layer containing $n_p = 256$ units. Every layer was equipped with the ReLU activation function, ensuring non-negativity. Notably, the output units of the network together form the representation that is used to compute the loss, i.e. $\mathbf{p}$. The weights of the feedforward network were all initialized according to an input size-dependent uniform distribution, following the PyTorch default [Paszke et al., 2019].

For the spatial objective (1), the input to the feedforward network consisted of minibatches of continuous Cartesian coordinates, sampled randomly and uniformly within a $2 \times 2$ square enclosure. For the conjunctive objective (2), the input to the network was a concatenation of randomly sampled Cartesian coordinates $\mathbf{x}$, and uniformly sampled scalar context signals $c$, i.e. $\mathbf{input} = \text{cat}(\mathbf{x}, c)$. Context signals were sampled uniformly in the interval $c \in [-2, 2]$. To increase the number of training samples, distances in either objective function were computed between minibatch elements.

The recurrent network consisted of a single vanilla recurrent layer equipped with the ReLU activation function, without added bias. Like the feedforward network, this network featured $n_p = 256$ recurrent units. In the purely spatial case, the recurrent state at a particular time $t$ was given by

$$\mathbf{p}_t = \text{ReLU}(W_{rec}\mathbf{p}_{t-1} + W\mathbf{v}_t),$$

where $W_{rec} \in \mathbb{R}^{n_p \times n_p}$ is a recurrent weight matrix, $W \in \mathbb{R}^{n_p \times 2}$ an input weight matrix, and $\mathbf{v}_t$ Cartesian velocity inputs. In the case of conjunctive encoding, the input at time $t$ was a concatenation of the velocity and a (time-constant) scalar context signal. Note that this also adds an additional column to the input weight matrix. To mitigate vanishing and exploding gradients, the recurrent weight matrix was initialized to the identity, similar to Le et al. [2015]. As with the feedforward network, losses were computed by comparing across trajectories (and contexts). The trajectory length was taken to be $T = 10$ timesteps. The initial recurrent state was computed by feeding the trajectory starting location (and optionally, the context) into a three-layer, densely connected network with 64, 64 and $n_p = 256$ units, respectively, each equipped with a ReLU activation function. We also trained a recurrent network on long sequences, without explicit positional information (position-independent state initialization, only velocity input) to demonstrate that our findings hold even when assumptions on model inputs are relaxed (see App. A for details).

For the recurrent network, inputs consisted of velocity vectors along trajectories in the same square region as for the feedforward case. Trajectories were generated by creating boundary-avoiding steps successively. A step was formed by sampling heading directions according to a von Mises distribution, alongside step sizes drawn from a Rayleigh distribution. If the step landed the trajectory outside the enclosure, the velocity component normal to the wall was reversed, bouncing the trajectory off the boundary. Initial trajectory steps were sampled uniformly and randomly within the square arena. The von Mises scale was taken to be $4\pi$, and the Rayleigh scale parameter $0.025$. At training time, data was created on the fly due to the low computational cost. All networks were trained for a total of 60000 training steps, with a batch size of 64. For each model, we used the Adam optimizer [Kingma and Ba, 2017] with a learning rate of $10^{-4}$. Unless otherwise specified, all models were trained with $\lambda = 0.1$, $\beta = 0.5$ and $\sigma = 0.25$. To quantify place field numbers and sizes for trained representations, we applied the thresholding- and connected area-approach used by Harland et al. [2021].

## 4.2 Spatial correlation & remapping

To evaluate representational changes in the face of changing context input, we ran the trained feedforward network on 32 linearly spaced contexts (in the range [-2, 2]). Following Leutgeb et al. [2004], we then computed the spatial correlation between unit ratemaps across contexts. Between-context spatial correlations were computed by correlating the ratemap of a unit in one context with the same unit's ratemap in another context in a binwise fashion. Ratemaps were only compared if a unit displayed non-zero activity in both contexts.

To investigate whether the recurrent network encoded lower-dimensional structures across different contexts, we employed Uniform Manifold Approximation and Projection (UMAP) [McInnes et al., 2018], with default parameters. We first created unit ratemaps using 50000 10-timestep trajectories, and a resolution of 32 bins (in both x- and y direction) for 5 linearly spaced context signals (in the range [-2, 2]). We then applied UMAP to the concatenated ratemaps and reduced the dimensionality of the ratemap population vector down to a two-dimensional space.

## 4.3 Orthogonal transformations

To explore how the spatial map of one context, $P_{c_1}$, can be transformed into the spatial map of another context, $P_{c_2}$, we applied orthogonal transformations. Our goal was to find an orthogonal transformation $Q$ that minimizes the Frobenius norm $\|\cdot\|_F$ between the transformed spatial map from context $c_1$ and the spatial map in context $c_2$. Here, $P \in \mathbb{R}^{n_p \times n_{bins}^2}$ and the number of spatial bins $n_{bins}$ was chosen to be 128. This problem, known as the Procrustes orthogonal problem [Schönemann, 1966], is defined by

$$\min_Q \|QP_{c_1} - P_{c_2}\|_F \qquad \text{s.t.} \quad Q^T Q = I. \tag{3}$$

Using the Singular Value Decompositon of $M = P_{c_2} P_{c_1}^T$, i.e. $M = U\Sigma V^T$, the orthogonal matrix $Q$ can then be computed as $Q = UV^T$.

To explore whether global orthogonal transformations could be used to form *new*, distinct representations (similar to global remapping), we transformed a learned representation using different global orthogonal transforms, and computed the spatial correlation between the original and transformed representations. Specifically, we considered random rotations, reflections and permutations as possible candidate transformations. To form $n_p$-dimensional rotation matrices, we used the *spe-*

*cial_ortho_group* from the SciPy library [Virtanen et al., 2020], while reflection matrices were formed by interchanging two columns of such a rotation matrix. Random permutation matrices were formed simply as permutations of the identity.

To determine whether orthogonal transformations could also be used to generate new representations *continuously*, we conducted a simple numerical experiment, wherein existing representations were transformed iteratively into previously unexplored regions (akin to an environment expansion). Specifically, we transformed population vectors at the end of an existing representation iteratively along the horizontal axis using the exponential map, i.e.

$$\mathbf{p}_{i+1} = e^{t_i S} \mathbf{p}_i, \tag{4}$$

where $S = \frac{1}{2}(A - A^T)$ ensures that $e^{tS}$ is orthogonal and $\mathbf{p}_0$ is a learned population vector. For simplicity, we take $A$ to be a randomly sampled permutation matrix. When extending the representation by a distance of $|\Delta\mathbf{x}|$ in one spatial dimension, we transform a particular $\mathbf{p}_i$ by setting

$$t_i = \sqrt{\frac{\ln((1-\beta)e^{-|\Delta\mathbf{x}|^2} + \beta)}{\mathbf{p}_i^T S^2 \mathbf{p}_i}},$$

which ensures that the similarity structure in the direction of transformation adheres approximately to (1); see App. C for details. We applied (4) iteratively to population vectors of the feedforward network starting at the boundary of the environment, extending the representation to twice its horizontal length. The corresponding step length was equal to the bin width for original ratemaps, $|\Delta\mathbf{x}| = 2/n_x$, with $n_x = 32$ being the number of bins in the horizontal direction.

## 4.4 Decoding

In our model, each output unit is presumed to encode a spatial location near its peak activity. To determine the decoded position at a specific location $\mathbf{x}$, we employed a weighted average of peak positions of the output units similar to Zhang et al. [1998]. The weights correspond to the activity levels of the respective units at that location. The decoded position $\hat{\mathbf{r}}(\mathbf{x})$ is thus given by:

$$\hat{\mathbf{r}}(\mathbf{x}) = \frac{\sum_{i=1}^n p_i(\mathbf{x})\mathbf{r}_i}{\sum_{i=1}^n p_i(\mathbf{x})}, \tag{5}$$

where $\hat{\mathbf{r}}(\mathbf{x})$ denotes the decoded position at $\mathbf{x}$, $p_i(\mathbf{x})$ the activity of unit $i$ at $\mathbf{x}$, and $\mathbf{r}_i$ indicates the peak location encoded by unit $i$. Notably, the decoding process does not incorporate all output units. Instead, units are prioritized based on their activity at position $\mathbf{x}$, and only the top $n$ most active units are included in the decoding.

As an alternative scheme that exploits the similarity structure of the learned representations, we implemented a simple population decoding procedure. This was done by generating and saving a "memory" of $M$ population vectors $\{\mathbf{p}_i^m\}_{i=1}^M$ and corresponding locations $\{\mathbf{r}_i^m\}_{i=1}^M$, and decoding subsequent locations as the location corresponding to the closest population vector in memory. For simplicity, we chose memory locations to be midpoints of unit ratemap bins, and memory population vectors the activity in that bin (for both feedforward and recurrent networks).

As a baseline, we also trained a linear decoder to predict Cartesian coordinates given network representations $\mathbf{p}$. Decoder weights were initialized according to a random uniform distribution, and trained using a batch size of 64, the Adam optimizer with a learning rate of $10^{-3}$, and 5000 (10000 for the RNN case) training steps. To compare decoding methods in the feedforward case, we chose a memory resolution of 32 bins (in both x- and y direction), and the same resolution to determine peak locations of units for Top n decoding. For the Top n scheme we used $n = 44$ based on the minimum decoding error across 10 models. The linear decoder was trained on population vectors of uniformly sampled locations. Finally, we evaluated all methods on a grid of 129 x 129 points. For the RNN, we used a memory resolution of 32 bins, and 5000, 10-timestep trajectories were used to create population ratemaps. Both methods were evaluated on 256 trajectories with 20 timesteps each.

## 4.5 Figures & code availability

Figures were created using BioRender.com, and code to reproduce all findings and figures is available at `https://github.com/bioAI-Oslo/ConjunctiveRepresentations`.

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

# Appendix / supplemental material

## A    Learning spatial representations without explicit position information

For both feedforward and recurrent models in this work, we make use of explicit Cartesian coordinates to train networks. In the real world, agents do not have access to exact coordinates labeling the environment, which could raise concerns about the transferability of our results to biological networks. We therefore train a recurrent neural network without explicit positional information, to show that our findings hold even when relaxing assumptions on available inputs.

Concretely, we consider, as before, an RNN with $n_p = 256$ recurrent units. For this network, however, we initialize the network using a trainable initial state that does not depend on data. We furthermore train this network on long sequences ($T = 500$ timesteps), and compute similarities over trajectory time, i.e. within a trajectory, rather than aggregating over multiple smaller trajectories. The input to the network is just Cartesian velocities along such trajectories, and similarities used for training are computed only using relative distances.

The results in Fig. A1 show that the model also learns place and band-like representations and is able to learn the objective (see Fig. A1 a), b) and d)), similar to our other findings. As with other networks, we can also decode positions from this network using either a simple linear decoder, or our population decoding scheme, demonstrating that the learned representations are decodable (Fig. A1c)). Here, we evaluate the model along the same 16 unseen long trajectories of 500 timesteps for both schemes. Training of the linear decoder was performed by repeatedly sampling population vectors using 500 timesteps-trajectories, for 5000 training steps using the Adam optimizer, a learning rate of $10^{-3}$, and a batch size of 64. Additionally, we used 100 trajectories of 500 timesteps to form the memory vectors with a resolution of 32x32 bins for the population decoding scheme (see Sec. 4.4). In both cases, we omitted the first 50 timesteps of the trajectories to avoid large initial errors caused by the network's lack of positional information.

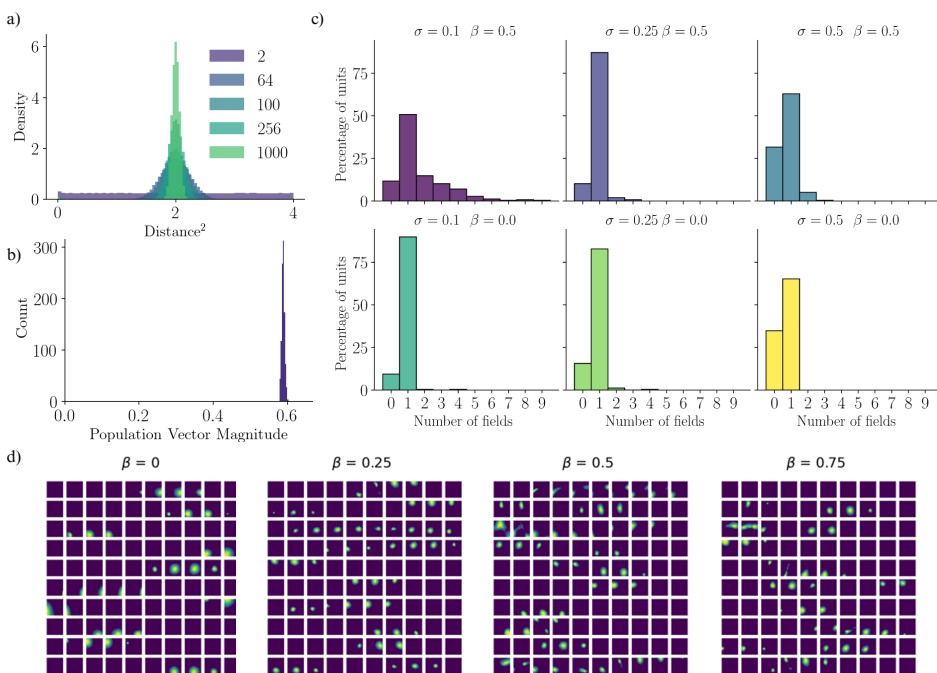

Figure A1: **Spatial representations without explicit position information**. a) Ratemap examples of randomly selected units of the long-sequence recurrent network. b) Training loss of the long-sequence recurrent network (data created on the fly). c) Mean decoding error of a linear decoder and the population decoding scheme on 16 unseen long trajectories. The dashed line indicates a naive case in which the decoded position is always at the center of the environment. d) Learned (left) and target (middle) similarity structure, alongside their difference (right) relative to center of arena, for the long-sequence recurrent network.

# B    Loss ablation and effects of $\beta$

We have previously noted that learned place field sizes are governed by the scale parameter $\sigma$. To further explore the influence of hyperparameters on learned representations, we performed an ablation study, wherein we train feedforward networks with $\lambda$ and $\beta$ ablated. We also explore the effects of changing the similarity measure, from Gaussian (depending on the square of the distance between locations/representations), to exponential in the Euclidean distance, i.e. $\mathrm{sim}(\mathbf{a}, \mathbf{b}) \propto e^{-|\mathbf{a}-\mathbf{b}|}$.

The results are shown in Fig. A2. With no activity regularization ($\lambda = 0$), units exhibit multiple smaller, but more irregular place fields (Fig. A2a)). Stronger population vector separation ($\beta = 0$) leads to highly unimodal representations, and larger place fields (Fig. A2b)). Notably, when both $\beta$ and $\lambda$ are set to zero (Fig. A2c)), units no longer display place-like tuning, but rather appear to change linearly across the arena, possibly reflecting the positional input to the feedforward network. Thus, place-like tuning in our model is dependent on either a non-zero similarity threshold, or activity regularization, but not both.

When similarities are computed using the Euclidean distance directly, (Fig. A2d)), we observe that network units still exhibit place-like spatial tuning. However, units now exhibit differing field sizes, with some large place-field units, and some units with smaller fields. Some units are also somewhat stripe-like. Thus, different similarity measures and distance functions can lead to different and possibly more expressive tuning profiles, which could be an interesting avenue for future investigation.

To better understand the role of $\beta$ in determining network representations, we first turn to an interesting fact of (normalized) vectors in higher dimensions; for large $n$, vectors on the $n$-sphere tend to reside at some intermediate distance from most other vectors; an example of this is shown in Fig. A3a), where distances are computed for random vectors of increasing dimension on the $n$-sphere. We therefore consider that dissimilarity may be meaningfully defined by some intermediate level of vector separation (as intermediately separated vectors are most common, and appear, in this sense effectively random).

Notably, this intuition can also be applied fairly directly to trained networks, as their population vectors tend to reside on the $n_p$-sphere (Fig. A3b), likely related to the applied L2 activity regularization. Being high-dimensional by design, network population vectors will thus tend to follow a similar distance distribution to Fig. A3a), again suggesting that a non-zero similarity threshold is sensible.

From a practical standpoint, a non-zero $\beta$ allows for more expressive representations, as representations may be reused across space and contexts, without incurring substantial loss. For example, if a unit fires at different, distant locations, this contributes to increased similarity. For $\beta = 0$ and sufficiently distant spatial locations, this situation would incur a loss, whereas a nonzero $\beta$ allows for some degree of similarity. Notably, even though states are of intermediate similarity, they can still be accurately decoded (as we demonstrate in e.g. Fig. 2i).

We indeed find that nonzero $\beta$ allows for more unit reuse across space. This can be seen in Fig. A3c), which demonstrates that higher values of $\beta$ (and smaller $\sigma$) lead to the emergence of units with multiple place fields (see e.g. the case where $\sigma = 0.1, \beta = 0.5$). We also observe a similar trend over different contexts, which is illustrated in Fig. A3d). In this case, units trained with $\beta = 0$ are only active around a narrow set of context values, whereas for $\beta > 0$, units are active across a wide range of distinct contexts, and place fields shift between contexts, similar to Hippocampal remapping.

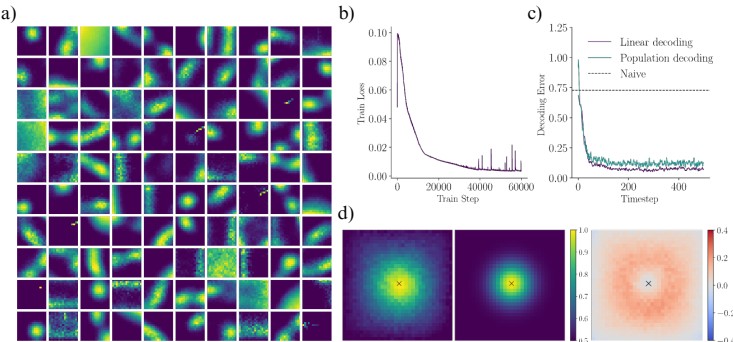

Figure A2: **Loss ablation and effect of similarity measure.** a) Ratemaps of randomly selected feedforward network units, when ablating $\lambda$. b) As in a), but for ablating $\beta$. c) As in a) and b), but for ablating both $\beta$ and $\lambda$. d) Ratemaps of trained feedforward units when the squared distance of the similarity measure is replaced by the Euclidean distance.

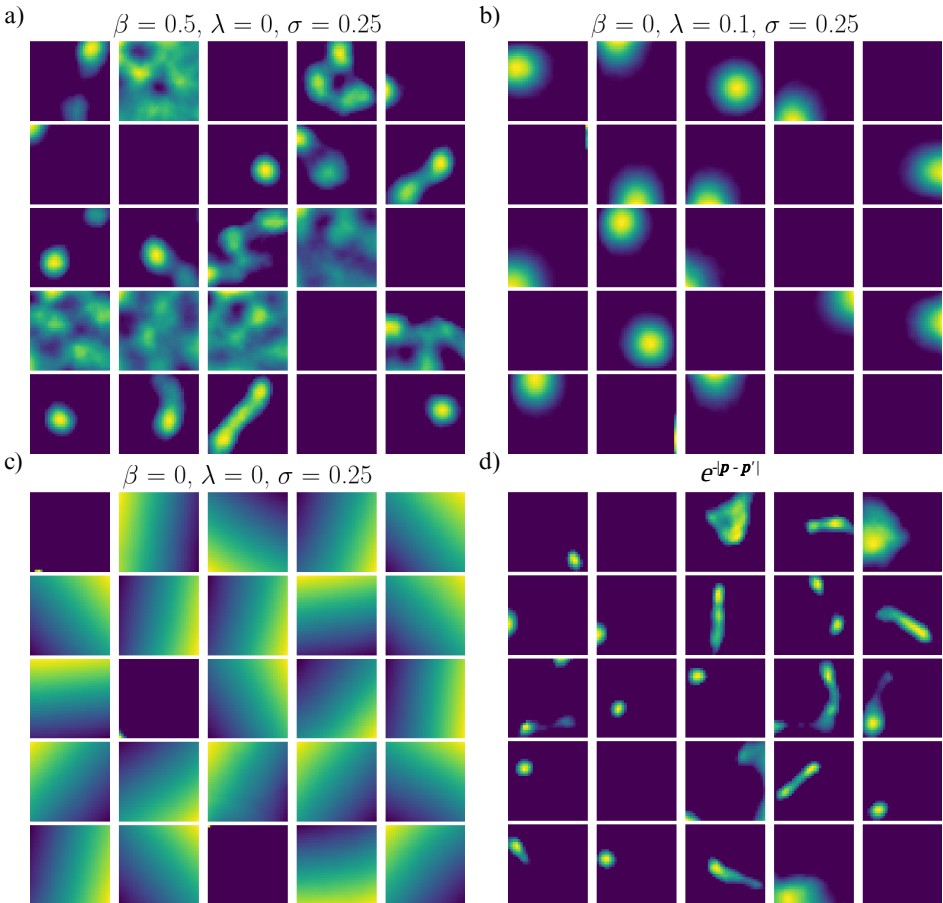

Figure A3: **Hyperdimensional computing and the effect of** $\beta$. a) Histogram of squared Euclidean distances between 512 randomly sampled vectors of different number of dimensions (legend) on the corresponding $n$-sphere. b) Distribution of population vector norms for a trained feedforward network with $\beta = 0.5$, $\sigma = 0.25$, and $\lambda = 0.1$. c) Histograms of the number of place fields for different parameter configurations (inset). d) Ratemaps of randomly selected units of a trained feedforward network with $\lambda = 0.1$, $\sigma = 0.25$, across different contexts for different values of $\beta$. For each value of $\beta$, one row represents one unit and each column one context value. Context values increase linearly from $-2$ (leftmost column) to 2 (rightmost column).

## C  Extending existing representations by orthogonal transformations

In this section we demonstrate how we can extend existing representations learned by the neural network to novel representations, using continuous orthogonal transformations. To do so, we employ the exponential map, given by

$$Q(t) = e^{tS},$$

where $t \in \mathbb{R}$ is a parameter, and we choose $S \in \mathbb{R}^{n_p \times n_p}$ to be a skew-symmetric matrix, which makes $Q(t)$ orthogonal. To extend an existing (purely) spatial representation, we need to transform it in a manner that respects (1). For orthogonal transformations that preserve norms, the quantity of interest is therefore distances in the neural representation before and after transformation. For orthogonal transformations, we have that

$$|\mathbf{p} - Q(t)\mathbf{p}|^2 = 2p^2 - 2\mathbf{p}^T Q\mathbf{p},$$

where $\mathbf{p}$ is the population vector we wish to extend from. Using the Taylor expansion of the exponential map

$$Q = I + tS + \frac{1}{2}t^2 S^2 + ...$$

and keeping terms up to second order, we have that

$$|\mathbf{p} - Q(t)\mathbf{p}|^2 \approx 2p^2 - 2(p^2 + t\mathbf{p}^T S\mathbf{p} + \frac{t^2}{2}\mathbf{p}^T S^2 \mathbf{p}),$$

and we can solve for the parameter $t$ approximately. Note that $S$ is skew-symmetric, so $\mathbf{p}^T S\mathbf{p} = 0$, and

$$t \approx \sqrt{-\frac{|\mathbf{p} - Q(t)\mathbf{p}|^2}{\mathbf{p}^T S^2 \mathbf{p}}}.$$

Notably, we can solve for the numerator term required to match the desired similarity structure in (1), i.e. demand

$$|\mathbf{p} - Q\mathbf{p}|^2 = -\ln((1-\beta)e^{-\frac{1}{2\sigma^2}|\mathbf{r}-\mathbf{r}'|} + \beta),$$

and insert into the expression for $t$, yielding

$$t \approx \sqrt{\frac{\ln((1-\beta)e^{-\frac{1}{2\sigma^2}|\mathbf{r}-\mathbf{r}'|} + \beta)}{\mathbf{p}^T S^2 \mathbf{p}}}.$$

We use this expression for the transformation parameter to extend a place-like representation horizontally into a region that has not been previously visited by the network, and demonstrate that the similarity objective is still (approximately) satisfied. Notably, this approach generalizes outside the training domain of the network that generated the starting representation without additional training. However, since we perform otherwise unconstrained transformations, representations are no longer guaranteed to be non-negative. One should also note that the above derivation does not ensure that the generated representations adhere to desired similarity structure in the vertical direction. However, we observe empirically that this is approximately the case (see similarity structure in Fig. 5e)), and hope to study the general case more closely in the future.

## D  Context-dependence of recurrent representations

Fig. A4 shows ratemaps for 10 randomly selected recurrent units of a network trained to minimize (2) during path integration evaluated across different contexts. As with the feedforward network, representations are place-like, and demonstrate multiple place fields, rate changes, and field shifts between contexts, similar to Hippocampal remapping.

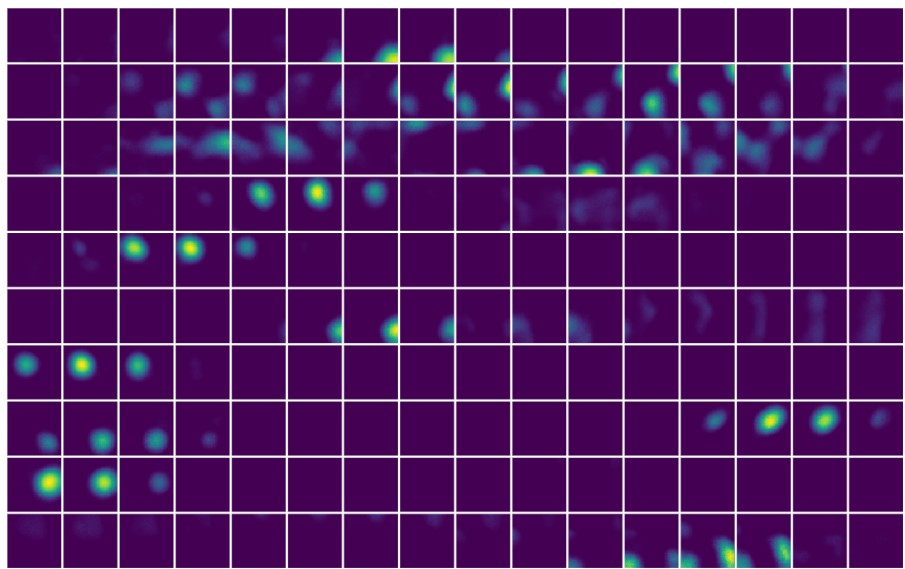

Figure A4: **Remapping Behavior in Recurrent Network**. Ratemaps of a recurrent network as a function of context, for a random selection of 10 units. Each row corresponds to one unit and each column to a particular context value. Contexts increase linearly from -2 (leftmost column) to 2 (rightmost column).

