# OpenReview forum: "Learning Place Cell Representations and Context-Dependent Remapping"
_NeurIPS.cc/2024/Conference — NeurIPS 2024 poster_

### Official Review · Reviewer_83Ga · 2024-07-02

**Soundness:** 3
**Presentation:** 3
**Contribution:** 3
**Rating:** 7
**Confidence:** 4

**Summary:**

This paper shows that tuning profiles similar to those of hippocampal place cells can emerge in neural networks trained to minimize a relatively minimalistic spatial encoding objective. In both a feedforward and recurrent neural network, the authors show that the trained network's units are tuned to one or a few spatial locations, similarly to place cells, and that they exhibit remapping-like phenomena when the context is changed. Interestingly, this remapping appears to be a continuous shift rather than a sudden switch. In the case of a recurrent neural network, the units' response profile resembles "band-like" activations that have also been observed in the hippocampal formation. They also show that the networks' learned spatial maps are rotation-equivariant, as it is possible to learn orthogonal matrices that transform the map in one context to that in another context.

**Strengths:**

- The paper elegantly focuses on a simple loss function, applied to standard neural network architectures. This makes the underlying assumptions clear, and helps to interpret the findings.
- The paper is clearly written, making the methods and analyses easy to understand.
- The analyses on the trained networks are thorough and creative.

**Weaknesses:**

- The title is vague and not very informative about the content of the paper. If possible, I would recommend changing it, to at least include a mention of hippocampal place cells, which are the main topic. The term "conjunctive" is also not particularly informative, as it suggests that the focus is on the binding between contextual information and spatial locations. Rather, context information as used here seems more like a tool to probe the networks' ability to maintain orthogonal spatial maps in different contexts. So perhaps a mention of "context-specific" spatial maps in the title would be more faithful to the paper's content.
- The paper misses a "related work" section. Of course, some related work is discussed in the introduction, but a more explicit and complete roadmap at the beginning would help, especially for non-expert readers, as the literature on the response profiles of hippocampal cells is extremely large and diverse. Throughout the paper, relevant empirical findings are cited in relation to the present results, e.g. band cells related to the RNN findings, or gradual remapping between contexts related to the context switching analysis (figure 3). Without a more thorough overview of relevant findings, however, it is hard to assess whether these response patterns would be expected to emerge in any model of the hippocampal formation, whether they are idiosyncratic findings that only appeared in a few studies, and similar questions. Adding this overview would help the reader to assess this work's significance and novelty.
- More generally, given the huge variety of response profiles that have been reported in the hippocampus, it would be helpful if the model's limitations with respect to empirical predictions were discussed more clearly. For example, how should the finding of gradual transition between contexts be interpreted? As the authors write, it is consistent with some empirical findings and inconsistent with others. Similarly, network units display peaks at multiple locations in space when the value of $\sigma$ is small. Is this dependence on the receptive field's size to be expected, given previous findings?
- The role of individual components of the loss function is not investigated (through ablation experiments), or at least discussed. In particular: (1) what is the role of the exponential (Gaussian) drop-off of the loss with spatial distance? Would simply plugging into the loss function e.g. the mean square error between the estimated and true position lead to similar results? One could argue that using the exponential drop-off in the loss function is a way to hard-code the place-cell-like response profile, and thus that this response profile is not a true emergent phenomenon. I am not saying that this is the case, but it would be good to add a discussion of the exponential's role, possibly with an ablation analysis showing its importance. (2) the use of a second-order loss is one of the main intuitions of the paper, as it displays several desirable properties which the authors discuss, such as rotation and translation invariance. However, the paper is missing an explicit comparison with a first-order objective whereby the network is trained to make absolute locations close together that are close to each other in space. What would happen with such a loss? How would the resulting response profile look? Explicitly testing this alternative loss could be a nice addition, but if this is not reasonable, an explicit discussion of the role of second-order similarity would be good. (3) the role of the lower bound $\beta$ is briefly mentioned, in relation to the hyperdimensional computing notion of nearly orthogonal vectors, but this analogy is not explored any further. Does near-orthogonality play a functional role here as well? Is it related to the separation between distinct contexts?
- In figures 2f, 4f and 5b, the difference between the learned spatial similarity structure and the objective is plotted, but this is not commented. What is the take-home message conveyed by these plots? Since the authors chose to show this information, a few words about what it might mean would be good.
- In figure 2g, the position decoding error (Euclidean distance) is plotted. However, it is hard to make sense of this measure without an upper bound. For example, what would be the expected error if before computing the weighted sum in (6) the units were shuffled? This is just an example of a possible error upper bound, and might not be the most pertinent, but comparing the empirical error to an upper bound would help evaluate the quality of the decoding.
- The recurrent neural network (RNN), besides having a different architecture, also receives input encoded in a different format, as velocity vectors rather than coordinates. I imagine that the purpose of this was to help it learn to path-integrate along the trajectories. However, this difference in input format makes it hard to determine whether the different response profiles (place vs. band) are due to the architecture or the input. Velocity vectors might already provide some minimal cues for the network to learn a more spatially distributed response. Would the response profiles of the RNN look the same if it received coordinates as input, like the feedforward network? And vice versa, would the feedforward network show more band-like responses if it received velocities?
- The band response profiles of the RNN seem to display less periodicity than those reported in Krupic et al. 2012. In that paper, the authors used a 2D Fourier analysis to determine the amount of periodicity in the band cells. Reproducing a similar analysis here would give some important insight into the significance of this finding and its relation to experimental data. Moreover, it would also be informative to report the proportion of units that displayed this kind of response profile. Was it close to that reported in Krupic et al.' s paper (44%)?

**Questions:**

- On page 4, line 120, the authors write: "distances in either objective function were computed between minibatch elements" does this mean that all possible pairs of datapoints were considered? If not, were they randomly sampled?
- It is not clear how the training procedure differs between the feedforward and recurrent models: if I understand correctly, the feedforward model received datapoints that were essentially random samples (batches of isolated x, y, c), while the recurrent model received temporally structured sequences. What was the purpose, then, of still sampling relatively smooth trajectories in training the feedforward model? Was the idea to have batches that are not IID, but somewhat "contextualized" (in each given batch, samples are correlated to each other) as in some continual learning experiments?
- Partially related to the previous question: what was the underlying "world model" behind using a randomly sampled context scalar at each datapoint? I would understand if input datapoints were just IID samples, in that case the idea would be that the same coordinates "mean" different things in different contexts, leading the network to learn separate spatial maps for the different contexts. But if there is some sequential structure in the data, does this correspond to a world in which an agent moves smoothly in space, but with a constantly varying context? And thereby, even visiting the same location repeatedly leads to a completely different experience? This seems counterintuitive given that contexts tend to be stable in the real world. I'm interested in what the reasoning was behind this choice, or perhaps I might have misunderstood the training procedure.

**Limitations:**

Limitations are properly acknowledged. The ones that have not been discussed are listed in the "weaknesses" section.

---

> ### Author Rebuttal · Authors · 2024-08-06
>
> We thank the reviewer for their positive and insightful comments.
>
> Thank you for your perspective on the title. As for the term conjunctive, we did initially focus on spatially bounded contexts, in addition to global ones. However, we ended up limiting the scope of this paper slightly, and we will revise the title to reflect this.
>
> Your point on relevant literature is a good one. We will, within space constraints, add an overview of relevant findings and models.
>
> On experimental findings:
> As you point out, it is difficult to manage all the (sometimes inconclusive) experimental findings, but we will definitely attempt to make this discussion more lucid. As for the gradual/sharp context transitions, our (possible) interpretation is that real-world contexts could be represented continuously, but that this only manifests under very controlled conditions, where the animal has access (and is attentive) to the environment. In our case, we feed in exact context signals, but in the future, it could be very interesting to let e.g. context inference be a part of the task, or adding uncertainties to the context signals.
>
> On the role of $\sigma$:
> In our model, $\sigma$ primarily governs the size of the place field (or, more specifically, the spatial scale at which population vectors should be similar). On the other hand, $\beta$ controls the similarity for distant points (both in space or context). We find that when *$\beta$* is somewhat large, the network is more likely to display multiple firing fields. However, as you point out, we also find that $\sigma$ influences the number of place subfields. A smaller $\sigma$ could alternatively be used to model place field behavior in larger environments. In these cases, multiple firing fields have been observed experimentally. See e.g. “Dorsal CA1 Hippocampal Place Cells Forms a Multi-Scale Representation of Megaspace” by Harland et al. (2021). However, in that work they observe multiple spatial scales in large spaces. How that relates to our work, could present an interesting avenue for future work.
>
> On loss ablation/evaluation:
> These are all excellent suggestions, which could strengthen our results. We will strive to do ablation studies, and compare with a first-order loss in the camera-ready version. Regarding your question; were you referring to the squared difference between distances (in representation and space)? If so, we would very much like to look into this simplified model in the future. We believe that decoding the position and computing the error between true and predicted locations might not lead to similar results, however, as it would likely require more constraints to avoid having the network learn a "trivial" Cartesian solutions. See e.g. Cueva and Wei (2018) (https://arxiv.org/abs/1803.07770) where they study this problem!
>
> On beta/hyperdimensional computing:
> We found that using $\beta > 0$ leads to the nice property that place cells and fields can be reused in different contexts. That is because reusing a place cell from context A in context B will make the population vectors of these two contexts somewhat similar. Essentially,
> this allows population vectors of different contexts to still have some non-zero similarity. Note that these (somewhat similar) population vectors can still be distinguished; they do not need be completely orthogonal, and can thus still be used for discriminating locations/contexts. We will motivate $\beta$ more clearly in the revised manuscript. See also Fig. 1 in the attached PDF.
>
> On figures 2f, 4f, 5b:
> We used those plots along with the loss in order to validate our model. We show that the error between the similarity structure imposed by the objective function and the actual learned similarity structure is low over space and that the model has indeed learned the objective. We refer to these figures briefly and will make this more clear in the revised manuscript.
>
> On decoding error:
> Thanks for pointing that out. We agree that it might be hard to interpret the errors without an upper bound. We therefore included a plot that shows exactly this in Fig. 5 (of the attached PDF) where we also show the decoding error for shuffled peak locations.
>
> On the use of different input models/inputs:
> You are right in that we did want the model to learn to path integrate and solve the same objective. We believe that this makes our model more biologically plausible as we do not rely on explicit position information, only on velocity inputs similar to speed (see [Kropff et al. 2015](https://www.nature.com/articles/nature14622)) and head direction cells (see [Muller et al. 1996](https://www.sciencedirect.com/science/article/pii/S0959438896800730?via%3Dihub)). Just to make sure we understand the question correctly: When you say training the feedforward on velocity signals do you have in mind providing the current position along with the velocity signal at each step, and predict the next state?
>
> On band representations:
> This is a good point; we can determine the periodicity of the learned bands and compare with experiments in time for the camera-ready version.
>
> Questions:
> On distances:
> Yes exactly, we computed all pairwise distances across minibatch elements (and timepoints in the RNN case).
>
> On training:
> In the feedforward case, all samples are IID both for the context and the spatial dimension. Here, we did not sample trajectories but sampled uniformly and continuously within the spatial and contextual range and then computed all pairwise distances. In the RNN case the context was constant along each trajectory but randomly sampled (uniformly and continuously) for each trajectory. Thank you for pointing that out. We will make this more clear in the camera-ready version.
>
> On context signals:
> See answer above. You are right, learning different spatial maps was exactly what we hoped for and therefore, we used IID samples. Along trajectories in the RNN case the context signal was constant. We will make sure to clarify this.

---

> > ### Comment · Reviewer_83Ga · 2024-08-13
> >
> > I thank the authors for their rebuttal. I believe all my concerns were addressed, including the addition of some informative analyses such as the role of $\beta$ in Fig. 1 of the attachment, and the upper bound in Fig. 5.
> >
> > In general, as the responses and additional analyses thus far mainly serve to clarify the points made in the paper, rather than changing it substantially, I will keep the same score.

---

> > > ### Author Response · Authors · 2024-08-13
> > >
> > > Thank you very much for your feedback and your valuable inputs.

---

### Official Review · Reviewer_qPN7 · 2024-07-09

**Soundness:** 3
**Presentation:** 3
**Contribution:** 2
**Rating:** 4
**Confidence:** 4

**Summary:**

This paper proposes a new model of hippocampus neurons by devising a new objective function that enforces particular distances between latent codes associated with different position inputs. A Gaussian distance weighting function is proposed. Experiments show that the resulting model produces hippocampus-like position selectivity in the context of learning from single observations or trajectories and demonstrates remapping-like change in selectivity of model units with drifting context values.

**Strengths:**

- The topic of modeling hippocampal representations is important.

- Paper is well written and easy to follow.

- Comprehensive experiments and analyses are conducted. Figures are also nicely made.

**Weaknesses:**

- My biggest issue with the proposed method is that it explicitly requires position information in its training and inference which is completely non-biologically plausible. The core of the problem of learning hippocampus like representation is doing so without access to position information. There are already some models that can learn such representations through predictive coding from observations and velocities alone (Whittington et al 2020 and 2021) or with relaxed assumptions (relative positions Schaeffer et al. 2023) which substantially diminishes the significance of the current work.

Whittington, J. C., Warren, J., & Behrens, T. E. (2021). Relating transformers to models and neural representations of the hippocampal formation. arXiv preprint arXiv:2112.04035.

- line 27: the preceding sentences speak of selectivity to spatial and non spatial features separately. Include refs for conjunctive coding in hippocampus.

e.g. in primates
Gulli, R.A., Duong, L.R., Corrigan, B.W. et al. Context-dependent representations of objects and space in the primate hippocampus during virtual navigation. Nat Neurosci 23, 103–112 (2020). https://doi.org/10.1038/s41593-019-0548-3

in rodents:
Anderson, M. I., & Jeffery, K. J. (2003). Heterogeneous modulation of place cell firing by changes in context. Journal of Neuroscience, 23(26), 8827-8835.


- A scalar context signal is used in all experiments. Can all contexts be represented as a scalar? what are the limitations associated with this assumption?

- decoding appraoch: I’m not familiar with this decoding approached and I don’t understand its implementation or logic from the description either. Why not using a simple linear readout to decode the position.

- section 3.4: more explanation is needed for motivating this analysis. maybe a copule of sentences at the beginning of section 3.4 since it is not clear why that is a useful property until all the results are presented

- section 3.4: what is the computational value of having the rotation invariance property? is learning an orthogonal transformation easier than learning the FF projection? Does it help with learning a task? Would an agent be able to learn tasks faster with that property? Putting this into the context of behavior would be helpful

**Questions:**

- why is there a $\beta$ term in equations 1 and 2? Looks like it's not needed

- line 265: "restricted orthogonal transformations could prove a viable way of modeling Hippocampal remapping." how would this approach be useful? Please explain.

**Limitations:**

Limitations were discussed

---

> ### Author Rebuttal · Authors · 2024-08-06
>
> We thank the reviewer very much for their insightful feedback. Below, we address some of the concerns and questions asked.
>
> We understand your concern about the biological plausibility of using Cartesian coordinates. It is true that in the FF case we use explicit position information. However, we used the FF model because of its simplicity and it is worth mentioning that our approach also works using only relative distances in the RNN case. Here, the input is just a velocity signal inspired by biological speed cells (see [Kropff et al. 2015](https://www.nature.com/articles/nature14622)) and head direction cells (see [Muller et al. 1996](https://www.sciencedirect.com/science/article/pii/S0959438896800730?via%3Dihub)). For the labels in the objective function we only need relative distances as well ($x_t-x'_t$). Lastly, while we initialize the RNN with the initial position in the current version, we show that a model trained on longer trajectories can also learn the objective without positional initialization. This version of our model does not use sensory information. We have added a figure to the appended PDF (see Fig. 6), showcasing how the RNN without any positional information does in fact learn similar representations as the networks used in the article. In conclusion, we can say that while in the simplified case we do use explicit position information, those are not necessary to learn the objective and arrive at the presented representations.
>
> We also understand the reviewer’s concern about the novelty of the learned spatial representations, and agree that interesting models with diverse cell types have been explored in other works. However, we would argue that our work is novel in several ways: For one, we focus specifically on the encoding properties that hippocampal representations should have, whereas most other works tend to focus more on grid cells and complex, joint models of the entorhinal-hippocampal circuit. Relatedly, our model is explainable, in the sense that learned representations can be understood directly from the objective function. Lastly, but perhaps our biggest novelty, is incorporating (possibly) non-spatial context information in the same model, and showing in an interpretable manner how Hippocampal remapping occurs as a consequence of this form of encoding. We would also highlight our finding that orthogonal transformations leave the objective invariant, as a novelty of this work.
>
> We thank the reviewer for pointing out that we should include references for conjunctive coding in Hippocampus. That's a very good point, and we will make sure to have those in place in the camera-ready version.
>
> The point about the context signal is very interesting. In our current version, we have always assumed a scalar signal for simplicity which might not be biologically plausible. It would mean that the entire context is encoded as a one-dimensional signal. However, our approach easily extends to higher-dimensional context signals. As we just use distances between two context signals (in the same way as we use distances in position, see equation 2 for reference) one can imagine the context to be encoded by more neurons in high-dimensional space. Likewise, one could also consider different forms of contexts signals, such as discrete ones.
>
> We regret that our description the decoding was unclear. Essentially, we use a weighted average of the peak locations of nearby place cells to decode position. The weight of each place cell is given by its activity in the particular location to be decoded, similar to e.g. [Zhang et al., 1998](https://journals.physiology.org/doi/full/10.1152/jn.1998.79.2.1017?rfr_dat=cr_pub++0pubmed&url_ver=Z39.88-2003&rfr_id=ori%3Arid%3Acrossref.org). For reference, we also trained a linear decoder and included the result in the attached PDF (Fig. 2).
>
> Regarding your comments on section 3.4: we will make sure to give a stronger motivation at the start of this section. We see an advantage in using orthogonal transformations as they preserve distances and leave the result of the objective function unchanged. So once we have found a representation that is a solution to the objective, we can form new representations using orthogonal transformation which are still guaranteed to be a good solution. In the context of behavior what this means is that we might just have to come up with a good representation of space once and then using orthogonal transformations we could form new spatial maps without having to relearn them from scratch. These maps could be used to represent different contexts. However, whether this manifests in terms of faster learning is something we want to explore further in the future.
>
> Questions:
> 1. You are right in that the $\beta$ parameter is not necessary for the basic idea of our approach. However, we found that using $\beta > 0$ leads to the nice property that place cells and fields can be reused in different contexts. That is because reusing a place cell from context A in context B will make the population vectors of these two contexts somewhat similar. Essentially, $\beta>0$ allows population vectors of different contexts to still have some non-zero similarity. Note that these (somewhat similar) population vectors can still be distinguished; they do not need be completely orthogonal, and can thus still be used for discriminating locations/contexts. We will motivate $\beta$ more clearly in the revised manuscript.
> 2. Mechanisms behind remapping are still poorly understood. We have shown that the distance preserving property is useful for the model we studied. From there, we believe there are some interesting questions that follow that could help in studying remapping : 1. Is there structure in the orthogonal transformations? 2. What (upstream) representations are needed to induce the orthogonal transformations in this manner? 3. Can this property also be shown experimentally?

---

> > ### Comment · Reviewer_qPN7 · 2024-08-12
> > **follow up comments**
> >
> > Thank you for considering my comments and responding to my questions. Several follow ups:
> >
> > > For the labels in the objective function we only need relative distances as well.
> >
> > To me computing the distance is almost the same as being position aware. Maybe one way to resolve this issue of biological implausibility is to show that the model can be trained with considering a proxy of distance like the number of steps travelled.
> >
> > > In conclusion, we can say that while in the simplified case we do use explicit position information, those are not necessary to learn the objective and arrive at the presented representations.
> >
> > I'm a bit confused which result is showing that access to explicit position information is not necessary.
> >
> > >  In the context of behavior what this means is that we might just have to come up with a good representation of space once and then using orthogonal transformations we could form new spatial maps without having to relearn them from scratch.
> >
> > I think this is potentially a very interesting claim and intuitive as well but it needs empirical validation. I suggest grounding this into an analysis where agent's behaviour is compared within a new environment when different parts of the model are fixed are left being trainable.

---

> ### Author Response · Authors · 2024-08-13
>
> Thank you very much for your insightful feedback.
>
> > To me computing the distance is almost the same as being position aware. Maybe one way to resolve this issue of biological implausibility is to show that the model can be trained with considering a proxy of distance like the number of steps travelled.
>
> We agree that the issue of how the brain might learn representations of relative distance is an intriguing area for future exploration. In our current work we wanted to focus on hippocampal place cells and remapping behavior using ML methods. To limit the scope we dit not want to get into the question of how the brain might learn representations of relative distance. However, we acknowledge that understanding how relative distance might be encoded is critical for a more complete model of spatial navigation. We just want to mention that existing literature such as the studies on grid cells serving as a distance metric, already provides some insights into how self-motion can be translated into relative distance representations (e.g., [Bush et al., 2015](https://www.cell.com/neuron/fulltext/S0896-6273(15)00628-5?_returnURL=https%3A%2F%2Flinkinghub.elsevier.com%2Fretrieve%2Fpii%2FS0896627315006285%3Fshowall%3Dtrue), [Stemmler et al., 2015](https://www.science.org/doi/10.1126/science.1500816), and [Dang et al., 2021](https://www.sciencedirect.com/science/article/pii/S0893608021001684?via%3Dihub)).
>
> > I'm a bit confused which result is showing that access to explicit position information is not necessary.
>
> We apologize for any confusion on this point. In Fig. 6 of the attached PDF we present the representations of a model that was trained without access to explicit euclidean coordinates, relying only on relative distances. These results demonstrate that the model can develop similar spatial representations as those trained with explicit position information (such as in the FF and RNN models with initial position initialization). This supports the idea that explicit position information is not necessary for learning the objective and arriving at the presented representations.
>
> > I think this is potentially a very interesting claim and intuitive as well but it needs empirical validation. I suggest grounding this into an analysis where agent's behaviour is compared within a new environment when different parts of the model are fixed are left being trainable.
>
> Thank you for this excellent suggestion. This could indeed provide empirical validation for the claim that a good representation of space could be used to form new spatial maps via orthogonal transformations, without relearning from scratch. We are excited about this direction and plan to explore and validate this idea further.
>
> We hope these clarifications address your concerns and thank you for your valuable input.

---

> > ### Comment · Reviewer_qPN7 · 2024-08-13
> > **keeping score**
> >
> > Thank you for clarifying my questions. Several of my questions were answered but there remains a few shortcomings some of which the authors acknowledge but cannot fully address within the time provided for rebuttal.
> >
> > I appreciate the authors trying to train their model with relative distance but to me the resulting model seems to be a major downgrade from the original one.
> >
> > I am sure that my comments about clarity of the manuscript and methods could be addressed properly in a revised version however I still feel that while the section on remapping with orthogonal transformations is potentially interesting, it needs further empirical controls to support the main claims, the details of which are mentioned in the comments.
> >
> > Given the above points, I can't convince myself to increase my score and vote for accepting the paper as is.

---

> > > ### Author Response · Authors · 2024-08-14
> > >
> > > Thank you for your feedback. We appreciate your recognition of the potential in our approach, especially the part on remapping with orthogonal transformations. We understand that further empirical controls are necessary to fully support our claims, and we will focus on addressing these points based on your comments. Additionally, we will ensure that the revisions to the methods and clarity of the manuscript are incorporated into the camera-ready version. We also value your constructive review and will work on addressing the concerns you've raised.

---

### Official Review · Reviewer_QBGM · 2024-07-09

**Soundness:** 3
**Presentation:** 3
**Contribution:** 3
**Rating:** 6
**Confidence:** 4

**Summary:**

This paper studies the emergence of “conjunctive representations” in the context of place cell representations. The authors investigate how their proposed similarity-based objective function produces context-dependent place cell representations that exhibit remapping behaviors across contexts. The proposed objective function further produces representations that are invariant to orthogonal transformations across contexts.

**Strengths:**

In general, I enjoyed this paper. It was well-written, thought-provoking, and for the most part, the results are clear. The results were not overstated, and the figures (for the most part) accurately conveyed the findings of the paper. The similarity-based objective function was also intuitive, and has sparked some considerations (personally) on how to implement such similarity-based objective functions in a self-supervised manner. I particularly found the UMAP representations visualized in Fig. 4g and h to be particularly helpful in conveying the primary claims of the paper.

The intuition provided of their loss function (lines 81-84) were particularly helpful.

**Weaknesses:**

1. The authors allude to this primary weakness in their limitations section, but it is worth mentioning again: The supervised nature of the task makes it difficult to understand how biologically/ecologically valid their proposed objective function is. In some ways, the formulation of the task is such that the agent already needs to have an ‘oracle’ understanding of the environment it will inhabit/traverse. Equation 2 requires quite a lot of supervised/labelled information.

2. It appears that the shapes of all environments (across contexts) are the same, i.e., grids of the same size. This makes it hard to infer how general the findings are across contexts with differently shaped grids. For example, would there still be 1-1 remapping (i.e., via some orthogonal transform) across contexts with differently shaped environments (e.g., square -> circle, or square of [2,2] size to [1,1] size)? It would have been interesting to see these differently shaped environments in the UMAP in Fig. 4g and h, and whether the environment-specific shapes would be preserved in that figure.

3. I found Figure 5 to be confusing. For example, what do panels 5c and d add? It is known that the product of QQ^T produces the identity, since that is the definition of an orthogonal transformation. If the goal of the figure is to convey that remapping occurs via orthogonal transformation, then it might be more helpful to just illustrate the norms of the two maps (across contexts) before and after the orthogonal transform. Perhaps as a control, it would be helpful to see what the norms would look like against some baseline, such as a random orthogonal matrix (or random matrix in general). I didn’t find the visualization of the orthogonal matrices themselves to be particularly helpful or intuitive.

**Questions:**

1. Minor point: One of the assumptions the authors include were that unit activations are bounded, yet the activation function they used was a ReLU (unbounded). Does this assumption actually matter in practice? Also, I think a reference for the assumptions of their loss function (lines 65-69) would be helpful.

2. Minor question: The authors mention that the spatial grid was 2x2 square enclosure. Initially, it read to me that this space was actually a 2x2 matrix, with 4 total points to sample from. But I assume that this space is sampled continuously within a grid, rather than just having four points in total.

3. Details on exactly how the loss was computed for a trajectory was unclear. For a given trajectory, which has 10 time points, was a distance metric computed between every pair of time points? Or was it just between the initial point from t_0 and t_k, for k=1,…,10, and then averaged across all pairwise losses?

4. Related to weakness #2 – do the authors think their results would hold across environments of different shapes?

5. Minor question: What is the rationale for only including the top n active units for decoding? I’m surprised that adding additional units would reduce the overall decodability of place. In principle, the decoder would only need to pay attention (ie have high coefs) for the units that are relevant, while ignoring other units.

6. Related to weakness # 3 – what is Fig. 5 trying to convey? I found the figure a bit obscure.

7. Another result that I think would have been nice to see (while bolstering the claims of the paper), is to see a continual learning experiment in the context of demonstrating that once place cells are found for one context, due to remapping, would producing place cells for a new context develop faster (i.e., with greater sample efficiency), since all that would be required to learn is the orthogonal transform?

**Limitations:**

1. As mentioned in other sections, the primary theoretical limitation is that it’s unclear how the objective function that discovers place cells (and context-dependent remappings) is biologically/ecologically valid, since it is highly supervised. An ideal objective function would be to discover one based on form of self-supervised loss.

2. Another primary limitation is the fact that all environments (across contexts) are the same shape, thus it is difficult to intuit whether the findings are generalizable across differently shaped environments.

---

> ### Author Rebuttal · Authors · 2024-08-06
>
> We thank the reviewer very much for their well-structured and insightful feedback. In the following, we want to address some of the concerns and answer the questions.
>
> Mentioned weaknesses
> 1. We agree that especially the use of explicit cartesian coordinates is biologically implausible. We just want to point out that in the RNN case we are able to use only relative distances in the input (velocity signal), and solve the same objective. Furthermore, for the ground truth labels, we also just need relative distances as opposed to actual euclidean coordinates (see the $x_t-x'_t$ term in the objective function). Internal signals of relative distance traveled have been shown to exist in the brain, for example in the form of speed cells (see [Kropff et al. 2015](https://www.nature.com/articles/nature14622)) and head direction cells (see [Muller et al. 1996](https://www.sciencedirect.com/science/article/pii/S0959438896800730?via%3Dihub)). Lastly, we want to mention that we can and have also trained an RNN model that does not require the initial location as an initialization but instead is initialized with a zero-state. We included a figure showing the resulting representations in the attached PDF (see Fig. 6). In conclusion, while in the naive feed forward case we do rely on implausible explicit positions, there are ways to make our model work with only relative distances which make it a lot more biologically plausible and leads to very similar representations. Note that this model can also easily be extended to accommedate sensory inputs.
> 2. That's a good point. In general, we can imagine two different cases here. First, when we use our model (for example the feed-forward one) as is and change the geometry. In that case, we would just sample from a different subset of cartesian coordinates and get similar representations to what we presented. We have included an example of a circular environment in the attached PDF (see Fig. 4). The second case would be to actually study how place cells morph / change in response to geometric changes which we believe is what the reviewer is referring to. In that case, the RNN model is more interesting and we could think about training it on one environment and then performing inference on a different geometry. Even though this was beyond the scope of this work, we agree with the reviewer in that it would be very interesting to see the changes in representations.
> 3. Thanks for pointing that out. Upon reviewing the figure we agree with the reviewer in that it would be useful to actually show the norm before and after applying the transformation. We will include that in the camera-ready version. Our idea behind showing the product $QQ^T$ was that the transformation is actually learned using the objective function shown in equation 5. Therefore, we wanted to validate that it is indeed orthogonal.
>
> Questions
> 1. That's a good point. We should have made more clear how our requirements are linked to the actual objective function. We will change that in the camera-ready version. The purpose of the ReLU is to make sure we only have non-negative activations (assumption 5). By bounding we here mean that the activity stays within a reasonable range, which is fullfilled by adding the regularization term in our loss function, making sure the model learns representations with low activity levels. We included this assumption since biological systems are generally energy efficient and cannot have unbounded activity. We did not use sigmoid because it tends to produce difficulties during training like vanishing gradients.
> 2. Yes, absolutely. We sampled uniformly and continuously from the grid (-1 to 1 in x and y direction), and did not just use 4 points.
> 3. Thanks for pointing that out. For one batch of trajectories $(batch size, timesteps, n)$ we flattened the batch size and timesteps dimension and then computed the pairwise distance between all points, so across time and trajectories, and averaged over all of those. We will make it more clear in the camera-ready version.
> 4. See point 2.
> 5. In this work, we used a non-trainable decoder, similar to [Zhang et al., 1998](https://journals.physiology.org/doi/full/10.1152/jn.1998.79.2.1017?rfr_dat=cr_pub++0pubmed&url_ver=Z39.88-2003&rfr_id=ori%3Arid%3Acrossref.org). However, you are right that for a trainable one, more units should not make the decoding worse. We tried this for the RNN, and the results are shown in the attached PDF (see Fig. 2). We will make this more clear in the camera-ready version.
>
> 6. The figure is trying to convey two main points:
> 	1. We can learn an orthogonal transformation using the described method (see learning objective in equation 2), apply it to our representations and get a different representation that still minimizes our objective in the same way (because distances are preserved). 2. We show that we can compute transformations that take you from one learned representation (in one context) to another. Thus, the behavior learned by the network can be summarized by an orthogonal transformation.
> 7. We absolutely agree and we are planning to look into continuous learning settings in the future. It is now already possible to just apply an orthogonal transformation to get new spatial representations (a new spatial map) right away where the similarity structure is (by definition) preserved. That's what we show in Fig 5c-e. In those cases, the spatial representations do not have to be learned but can just be computed using the transformation. After showing that, one can imagine that we could learn one initial representation and then just produce new ones using orthogonal transformations instead of re-learning them. Relatedly, we identified two interesting questions which we also plan to study further in future work: 1. Is there structure in the orthogonal transformations? 2. What upstream representations are needed to induce the orthogonal transformations in this manner?

---

> > ### Comment · Reviewer_QBGM · 2024-08-09
> > **Reviewer response to author rebuttal**
> >
> > I thank the authors for engaging with my questions and providing clarifying answers.
> >
> > In general, I am supportive of this paper, though maintain that I find the results limited in terms of evaluation (e.g., fixed environment with 1-1 mappings across environments).
> >
> > As a comment -- I do agree with one of the other reviewers that I think the current title is uninformative, and that it would be helpful to make it more specific.

---

> > > ### Author Response · Authors · 2024-08-13
> > >
> > > Thank you very much for your feedback and valuable input.
> > >
> > > We agree with the reviewers' suggestions and will ensure that we use a more informative title in the camera-ready version. Additionally, we appreciate your ideas regarding changes in the geometry of the environments. We are planning to further explore the concept of training our RNN model in one geometry and then studying the changes in representations in response to alterations in geometry. We believe that this approach can further validate our model and provide valuable insights. Thank you for your suggestions.

---

### Official Review · Reviewer_mSmA · 2024-07-12

**Soundness:** 2
**Presentation:** 3
**Contribution:** 3
**Rating:** 5
**Confidence:** 3

**Summary:**

The paper explores neural network models that mimic hippocampal place cells by learning spatial and contextual representations through a similarity-based objective function. It demonstrates how both feedforward and recurrent neural networks can encode and remap these representations in response to context changes, akin to biological place cells. The study also examines the application of orthogonal transformations to generate new spatial maps from learned representations without retraining.

**Strengths:**

1. The objective function proposed in this paper, aimed at maintaining similarity between representational and physical spaces, is both intuitive and persuasive. It successfully achieves localized representation while enabling the network to have remapping capabilities.
2. The idea of using orthogonal transformations for remapping is intriguing and insightful.
3. The authors have a clear understanding of the model's limitations and have discussed them in detail.

**Weaknesses:**

1. The paper's claim of 'conjunctive representation' seems somewhat overstated. In my view, conjunctive representation should involve a joint representation of location and localized sensory stimuli, whereas this paper merely represents context as a new dimension, enabling remapping under different contexts. I believe that remapping is not equivalent to conjunctive coding.
2. In the experimental setup of this paper, the shape of the environment remains unchanged (2x2), with only the context cues changing. For real animal experiments, would such a setup typically lead to rate remapping instead of global remapping?
3. The core of the proposed objective function is to maintain similarity before and after representation, similar to UMAP, which aims to preserve geometric features of data points before and after dimension reduction. Using UMAP to validate the model is not very convincing. Have the authors validated their model using other dimensionality reduction methods?
4. If, as the authors speculate, place cells remapped through orthogonal transformations, one would expect to observe no change in the pairwise correlations between neurons before and after remapping, which seems not to have been observed experimentally.
5. As the authors themselves discuss, the approach proposed in the paper still lacks biological plausibility.

**Questions:**

Please refer to weaknesses 2,3

**Limitations:**

The authors have adequately discussed limitations of their work.

---

> ### Author Rebuttal · Authors · 2024-08-06
>
> We thank the reviewer very much for their well-structured and fair feedback. In the following, we want to address some of the concerns and answer the questions that arose.
>
> Regarding the listed weaknesses we have a few comments that we hope might clarify a few things and answer some of the questions asked by the reviewer.
> 1. We absolutely agree that remapping and conjunctive coding are not the same thing. In the current version of the article, we only use a global context signal and show how in this simple case the conjunctive encoding of space and global context can lead to remapping. One can, however, also imagine a case in which the context signal itself depends on the location ($c(x)$) and then becomes a localized context. From that definition, one can derive many interesting cases and experiments which we plan to further investigate in the future. We understand, however, how this part about the conjunctive coding might be unclear and will make sure to make this more clear in the camera-ready version.
> 2. It's true that global remapping is usually associated with a change in the geometry and rate remapping with contextual changes. However, for rate remapping, the contextual changes are typically rather small. For bigger contextual changes (such as smell and room color), partial remapping has been observed which includes the relocation of place fields of cells (see [Latuske et al. 2018](https://www.frontiersin.org/journals/behavioral-neuroscience/articles/10.3389/fnbeh.2017.00253/full) for a good overview on those different types of remapping). An example can be found in [Leutgeb S. et al. (2005)](https://www.frontiersin.org/journals/behavioral-neuroscience/articles/10.3389/fnbeh.2017.00253/full#B40 "https://www.frontiersin.org/journals/behavioral-neuroscience/articles/10.3389/fnbeh.2017.00253/full#b40"). Here, they observe partial and even global remapping when switching from one room to another but keeping the enclosure constant (hence, no change in geometry). Overall, there are different results from different labs making it hard to say what exactly the requirement is for at least partial remapping (remapping that includes change in the position) to happen. We plan to to further discuss that in our article. Thank you very much for pointing that out.
> 3. The purpose of using UMAP was to visualize the learned similarity structure. We aimed to validate the models learned representations by showing the following:
> 	1. The loss saturates at a low value (showing that the objective was learned). It is also worth mentioning that the data is generated on the fly (we are not using a pre-created dataset) so it is basically a test loss.
> 	2. The learned representations are a good representation of space as the location can be decoded.
> 	3. The representations are place-like.
>   We have included a further validation using PCA in the attached PDF (see Fig. 3). Here, we show that while the dimensionality for each spatial map (one for each context) is roughly the same (90% explained variance with around 8 components), while the dimensionality for all spatial maps combined is much higher (<50% of variance explained with 12 components). Therefore, the spatial maps for different contexts have different low-dimensional features and are indeed different.
> 4. You are right, and we agree that one would not expect to see any pairwise correlations between neurons. In Fig. 3b we show that for the learned representations of different contexts the place fields of neurons are indeed uncorrelated between contexts, as expected. When we talk about rotations (or orthogonal transformations) later what we mean is the rotation of the entire population vectors. In that case, the pairwise correlations (before and after) of single neurons are not preserved. In fact, one could use a full permutation of the population vector as special case of an orthogonal transformation. In that case, since the place fields of different place cells are in different locations, the neurons' place fields would be uncorrelated. We hope this answers the question, and we are of course happy to provide more details on how we went about the rotations if anything is still unclear.
> 5. We agree that especially the use of explicit cartesian coordinates is biologically implausible. We just want to point out that in the RNN case we are able to use only relative distances in the input (velocity signal). Furthermore, for the ground truth labels, we also just need relative distances as opposed to actual euclidean coordinates (see the $x_t-x'_t$ term in the objective function). Internal signals of relative distance traveled have been shown to exist in the brain, for example in the form of speed cells (see [Kropff et al. 2015](https://www.nature.com/articles/nature14622)) and head direction cells (see [Muller et al. 1996](https://www.sciencedirect.com/science/article/pii/S0959438896800730?via%3Dihub)). Lastly, we want to mention that we can and have also trained an RNN model that does not require the initial location as an initialization but instead is initialized with a zero-state. We included a figure showing the resulting representations in the attached PDF (see Fig. 6). In conclusion, while in the naive feed forward case we do rely on implausible explicit positions, there are ways to make our model work with only relative distances which make it a lot more biologically plausible and leads to similar representations. These models solve the same objective function given in equation 2.

---

### Official Review · Reviewer_wYa3 · 2024-07-12

**Soundness:** 2
**Presentation:** 3
**Contribution:** 2
**Rating:** 5
**Confidence:** 3

**Summary:**

How do place cell responses to location and context emerge? The authors propose that place cell response emerge from a similarity-based object function that encourages places that are closer in space to be encoded as more similar representations. Specifically, the authors find that their model learns place-like representations (operationally defined here as the peak activation locations spanning the training arena and being able to decode position from the model's output). They also find that, when the model and objective function are extended to include context, the model's rate maps change as a function of context, and these changes in the model's representations (due to context changes) are orthogonal transformations.

**Strengths:**

Overall, I applaud the effort to use machine learning techniques as task-performing models of neural activity.

The paper offers an interesting perspective on remapping representations based on context information. The idea that remapping can be accomplished by orthogonal transformations could also be applied in future work to other types of neural networks and/or modeling different brain regions.

The writing is clear and well-organized. The figures are excellent, well-made, and easy to understand.

**Weaknesses:**

- My major concern is that the input to the feedforward network is the ground truth location (the animal's position as Cartesian coordinates). It's not particularly impressive that a model that directly receives position information is able to encode position information, nor that we can decode the position from its representation. If their main research question is "how place cell responses emerge" it doesn't make sense to feed them place information directly.

- Beyond the potential redundancy issue of having place information be both the input to and output from the models, the bigger issue is that their model does not mimic the task that the brain (or place cells more specifically) need to learn. The brain needs to figure out location through its sensory data without being given veridical Cartesian coordinates. The recurrent model (which only has access to velocity information) is much more interesting, but it only used in one out of 4 experiments.

- The authors describe their objective function as "a minimal, similarity-based objective," but it doesn't seem particularly minimal. Their objective function include five different constraints (page 2, lines 65-69), not including the version for incorporating context information. It would be helpful for the authors to spend more space in the paper describing their objective function in a way that highlights its simplicity.

**Questions:**

Q1. Why are there different inputs given to the feedforward vs. recurrent models?
Q2. Relatedly, does the objective function get modified for the recurrent model (since its inputs are velocity rather than the position coordinates)?
Q3. If the feedforward model (without context) only receives Cartesian coordinates as inputs (input dimensionality of 2), why does the model need multiple layers with 64 and 128 hidden units, respectively? In the world of machine learning, the model is quite small, but for only encoding 2 numbers, it might be over-parameterized.
Q4: What is the difference between Fig 4g and 4h?

**Limitations:**

See weaknesses.

---

> ### Author Rebuttal · Authors · 2024-08-06
>
> We thank the reviewer very much for their well-structured and insightful feedback. In the following, we want to address some of the concerns and answer the questions.
>
> We absolutely understand the concern about the access to euclidean position in the feedforward model. Here, the RNN model offers a variant where explicit positions in the input are not needed. We show in Fig. 4 that we get very similar representations with this model suggesting that the model can learn to path integrate and solve the objective using only relative displacements. Such signals have already shown to exist in MEC, for example in the form of speed cells (see [Kropff et al. 2015](https://www.nature.com/articles/nature14622)) and head direction cells (see [Muller et al. 1996](https://www.sciencedirect.com/science/article/pii/S0959438896800730?via%3Dihub)). Even for the labels in the objective function one could simply use the relative distance ($x_t-x'_t$) instead of the explicit cartesian coordinates. The choice to use those was mainly to keep things simple. Again, the above mentioned internal signals could provide this information. In addition to the results presented in the manuscript, we have included results for a variant of our RNN model in the attached PDF (see Fig. 6). This model does not even need access to the initial position and therefore can be trained and used without any explicit coordinates, relying only on relative distances. We believe that this makes our model much more biologically plausible as this is, as you say, closer to the problem the brain has to solve.
>
> Your comment on the simplicity of the objective function is a very good point. We are planning to spend more time on making the purpose of the different terms more clear in the camera-ready version. To us, the simplicity lies in the fact that basically we are just saying similar locations (or contexts) in the outside world should also be similar in neural space, as captured in equation (2).
>
> It is worth mentioning that the position and the context signal are incorporated in the loss in the very same manner. To prove our main point, the exponential terms alone are enough. The regularization term helps ensure rather small activations, while the $\beta$ parameter allows our neural representation to keep a certain similarity even for very distant physical locations. This leads to the nice property that neurons can be reused (as reusing the same neuron in a different context always leads to a certain similarity).
>
> Answer to the questions.
> 1. We are feeding in different inputs in order to show that the model does not necessarily need explicit euclidean coordinates but can also rely on relative distances (velocity) and path integrate those to solve the problem of representing close physical point similarly and far points dissimilarly, hence solving the same objective function. By doing so we are trying to provide a more biologically plausible model that does not need the ground truth location. In the new version we attempt to make the motivation behind this more clear.
> 2. The objective function is the same for both models. So even though the inputs are velocities in the RNN case, the model still has to solve the problem of representing similar euclidean coordinates similarly and dissimilar ones dissimilarly. The only labels required are distances and the initial position. However, as we show in Fig. 6 (in the additional figure panel provided along with this rebuttal), when using longer trajectories our model can even solve the problem without access to the initial position, and therefore does not need any explicit position information.
> 3. That's a very good question. Based on the theory on deep neural networks acting as universal function approximators, we wanted to make sure our network is deep and expressive enough. You are absolutely right in that our network is over parameterized. Here, our goal was not to come up with a low-dimensional solution but instead have the neural network learn a representation similar to those of Hippocampal cells.
> 4. Thank you for pointing out the lack of clarity in this figure. In Fig 4g we are showing multiple single trajectories in the UMAP space while in Fig. 4h we sampled from a lot of trajectories and averaged the binned activity over positions. We will make sure this is more clear in the camera-ready version.

---

> ### Comment · Reviewer_wYa3 · 2024-08-12
>
> Thanks to the authors for their thorough and clear reply! The additional results in the rebuttal attachment are particularly convincing to me. If the authors are going to include this in their camera-ready version (either in the main manuscript or the appendix), then I would like to change my score. Can the authors confirm that they will be adding these new analyses?

---

> > ### Comment · Reviewer_wYa3 · 2024-08-13
> >
> > Given that we're on the last evening before the reviewer response period is over, I'm going to go ahead and raise my score on the assumption that the authors will include the additional results in the rebuttal attachment in their camera ready. (My comment was originally not visible to the authors and I only recently realized that and updated it, so they likely would not have time to respond.)

---

> > > ### Author Response · Authors · 2024-08-14
> > >
> > > Thank you for your positive feedback and for considering an increase in your score based on our additional analyses. We are pleased to confirm that we will include these new results into the camera-ready version of our manuscript. We appreciate your valuable input.

---

### Author Rebuttal · Authors · 2024-08-06

We would like to sincerely thank for their positive and knowledgable feedback. We attached a PDF including supplementary figures to address some of your major concerns. Those figures are referenced in our individual rebuttals. Notably, we have included ratemaps of an RNN model that has been trained on only relative distance, without initializing it with the initial euclidean position. In this case, we do not need explicit coordinates, making it more biologically plausible. Note that this model can also easily be extended to accommedate sensory inputs. We believe that including your suggestions in our revised camera-ready version will strengthen manuscript even further. We hope we were able to address all questions and concerns in a satisfactory manner. Of course, we are happy to discuss any additional points or answer questions that you might still have.

---

### Decision · Program_Chairs · 2024-09-25

**Decision:**

Accept (poster)

**Comment:**

This paper introduces a novel objective function for modeling spatial representation akin to that of the hippocampus by forcing the spatially close positions to have similar latent representations. Place cell-like representations emerged in the model and context information was encoded as an orthogonal transformation within the same representational space.

The reviewers unanimously acknowledged the intriguing ideas presented in the paper, particularly the encoding of context information as an orthogonal transformation. However, several major concerns were raised by most of the reviewers:
1) Position information as input: A key issue highlighted by some of the reviewers is that the position information is provided as input to the model, making it unsurprising that the network develops a representation of this information. This is a significant criticism. The authors addressed this by including results that use relative distance instead of absolute position information. However, this solution divided the reviewers, with some agreeing and others disagreeing that it adequately addresses the criticism.

2) Remapping vs. conjunctive coding: Two reviewers pointed out that the conjunctive coding discussed in the paper is not the same as what is commonly understood in the field. They suggested changing the title to better reflect the content of the work. The authors agreed to use a more informative title in the camera-ready version.

3) Limited environment geometry: The use of only one environment geometry raises concerns about the generalizability of the results across different contexts and grid shapes.

While the paper offers several novel contributions, the reviewers noted that the claims require further experiments and controls to be fully substantiated. That being said, given that the majority of reviewers leaned toward acceptance, I recommend this paper be accepted for a poster presentation. I strongly encourage readers to carefully review the discussion between the reviewers and the authors.